# WorldPlay: Towards Long-Term Geometric Consistency for Real-Time Interactive World Modeling

Wenqiang Sun [* 1 3]   Haiyu Zhang [* 2 3]   Haoyuan Wang [* 3]   Junta Wu [3]   Zehan Wang [3]   Zhenwei Wang [3]
Yunhong Wang [2]   Jun Zhang [1]   Tengfei Wang [3]   Chunchao Guo [3]

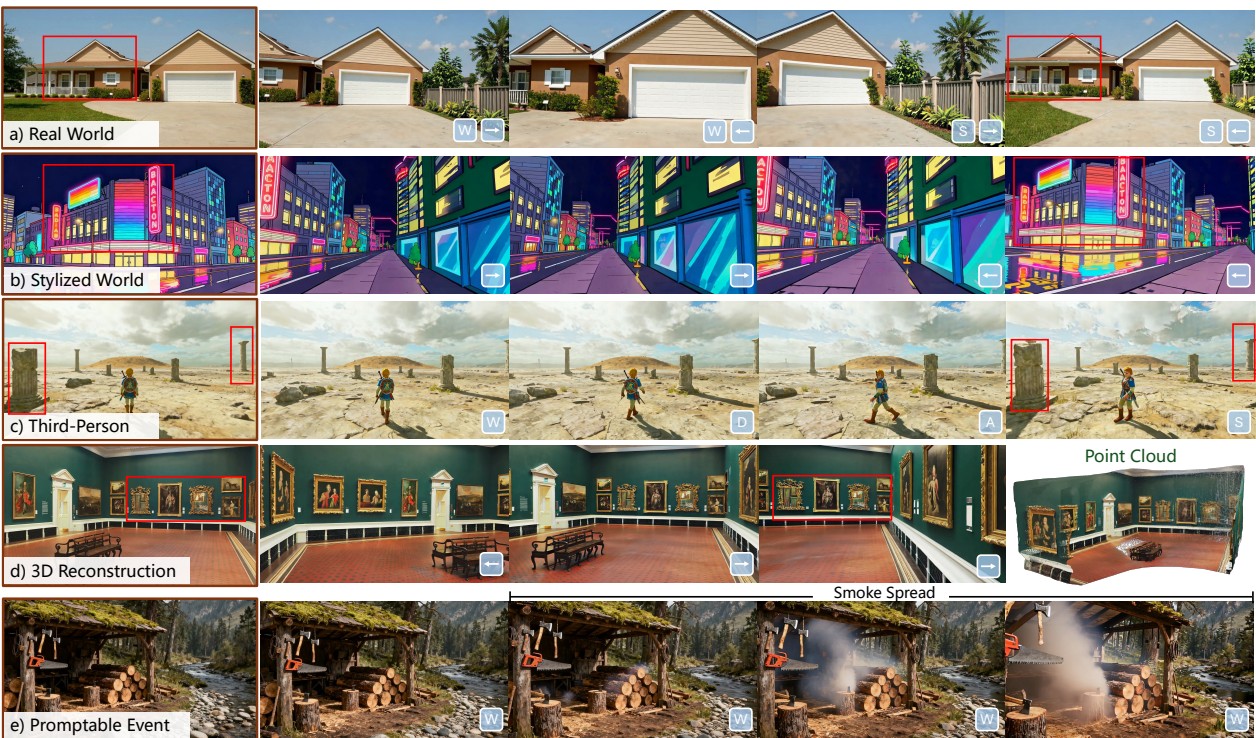

*Figure 1.* **WorldPlay is a real-time, interactive world model that achieves long-term geometric consistency.** It responds to user navigation commands in a streaming fashion, while maintaining scenes remain coherent when revisiting *(shown in red boxes)*. Our model shows remarkable generalization across diverse scenes, including **(a)** real world, **(b)** stylized world, and **(c)** third-person agent control. Furthermore, it supports **(d)** 3D scene generation via reconstruction and **(e)** dynamic world events triggered by text-based manipulation.

## Abstract

This paper presents WorldPlay, a streaming video diffusion model that enables real-time, interactive world modeling with long-term geometric consistency, resolving the trade-off between speed and memory that limits current methods. WorldPlay draws power from three key ingredients. 1) We use a Dual Action Representation to enable robust action control in response to the user's keyboard and mouse inputs. 2) To enforce long-term consistency, our Reconstituted Context Memory dynamically rebuilds context from past frames and uses temporal reframing to keep geometrically important but long-past frames accessible, effectively alleviating memory attenuation. 3) We also propose Context Forcing, a novel distillation method designed for memory-aware model. Aligning memory context between the teacher and student preserves the student's capacity to use long-range information, enabling real-time speeds while

*Equal contribution [1]Hong Kong University of Science and Technology [2]Beihang University [3]Tencent Hunyuan. Correspondence to: Jun Zhang <eejzhang@ust.hk>, Tengfei Wang <tengfeiwang12@gmail.com>, Chuncaho Guo <chunchaoguo@gmail.com>.

*Proceedings of the 43rd International Conference on Machine Learning*, Seoul, South Korea. PMLR 306, 2026. Copyright 2026 by the author(s).

preventing error drift. Taken together, WorldPlay generates long-horizon streaming 720p video at 24 FPS with superior consistency, comparing favorably with existing techniques and showing strong generalization across diverse scenes. Project page and online demo can be found: https://3d-models.hunyuan.tencent.com/world/ and https://3d.hunyuan.tencent.com/sceneTo3D.

## 1. Introduction

World models are driving a pivotal shift in computational intelligence, moving beyond language-centric tasks towards visual and spatial reasoning. By simulating dynamic 3D environments, these models empower agents to perceive and interact with complex surroundings, opening up new possibilities for embodied robotics and game development.

At the forefront of world modeling is real-time interactive video generation, which aims at autoregressively predicting future video frames (or *chunks*) to deliver instant visual feedback in response to every user's keyboard command. Despite significant progress, a fundamental challenge persists: *how to simultaneously achieve real-time generation (speed) and long-term geometric consistency (memory)* in interactive world modeling. One class of methods (Decart, 2024; Parker-Holder et al., 2024; He et al., 2025b) prioritizes speed with distillation but neglects memory, resulting in inconsistency where scenes change upon revisit. The other class preserves consistency with explicit (Li et al., 2025b; Ren et al., 2025) or implicit (Xiao et al., 2026; Yu et al., 2025b; Chen et al., 2025) memory, but complex memory makes distillation non-trivial (Sec. 3.4). As summarized in Table 1, the simultaneous achievement of both low latency and high consistency remains an open problem.

To tackle this challenge, we develop **WorldPlay**, a *real-time and long-term consistent world model for general scenes*. We consider this problem as a next chunk (16 frames) prediction task for generating streaming videos conditioned on actions from users. Building upon autoregressive diffusion models, WorldPlay draws power from the model's three key ingredients below.

The first is **Dual Action Representation** for control over agent and camera movement. Previous works (Decart, 2024; Parker-Holder et al., 2024; He et al., 2025b) typically rely on discrete keyboard inputs (*e.g.*, W, A, S, D) as action signals, which afford plausible, scale-adaptive movement but suffer from ambiguity for memory retrieval that requires revisiting exact locations. Conversely, continuous camera poses $(R, T)$ provide spatial locations but cause training instability due to scene scale variance in training data. To combine the best of both worlds, we convert action signals into continuous camera poses and discrete keys, achieving

robust control and accurate location caching.

The second key design is **Reconstituted Context Memory** for maintaining long-term geometric consistency. We actively reconstitute the memory through a two-stage process, moving beyond simple retrieval (Yu et al., 2025b; Xiao et al., 2026). It first dynamically rebuilds a context set by querying past frames based on spatial and temporal proximity. To overcome the long-range decay (the fading influence of distant tokens in Transformers (Su et al., 2024)), we propose *temporal reframing* to rewrite positional embeddings of these retrieved frames. This operation effectively "pulls" geometrically important but long-past memories closer in time, forcing the model to treat them as recent. This process keeps the influence of relevant long-range information preserved, enabling robust free extrapolation with strong geometric consistency.

The final key ingredient is **Context Forcing**, a novel distillation method designed for memory-aware models to enable real-time generation. Existing distillation methods (Chen et al., 2024a; Huang et al., 2025c; Yin et al., 2024a) fail to keep long-term memory as there is a fundamental distribution mismatch: training a memory-aware autoregressive student to mimic a memory-less bidirectional teacher. Even when augmenting teacher with memory, mismatched memory context will cause distribution diverge. We solve this by aligning the memory context for teacher and student during distillation. This alignment facilitates effective distribution matching, enabling real-time speed without eroding the memory while alleviating error accumulation over long sequences.

Taken together, WorldPlay achieves real-time, interactive video generation at 24 FPS (720p) while maintaining long-term geometric consistency under streaming user control. The model is built on a large-scale, curated dataset of 320K real and synthetic videos with a custom rendering and processing platform. As shown in Fig. 1, WorldPlay shows superior generation quality and remarkable generalization across diverse scenes including first- and third-person real and stylized worlds, and supports applications ranging from 3D reconstruction and promptable events.

## 2. Related Work

**Video Generation.** Diffusion models (Ho et al., 2020; Lipman et al., 2023; Song et al., 2021) have emerged as the state-of-the-art approach in video generative modeling. (Chen et al., 2024b; Guo et al., 2024; Yang et al., 2024) adopt the latent diffusion model (LDM) (Rombach et al., 2022) to learn video distribution in the latent space, achieving efficient video generation. Recently, autoregressive video generation models (Chen et al., 2024a; Henschel et al., 2025; Kim et al., 2024) theoretically enable one to generate unlim-

*Table 1.* Comparison with recent interactive world models. WorldPlay distinguishes itself as a general-domain model that simultaneously achieves long-horizon video generation, flexible action control, real-time interactivity, and long-term geometric consistency. 'Con.' and 'Dis.' represent continue and discrete action, respectively.

| | Yume (Mao et al., 2025b) | Matrix-Game 2.0 (He et al., 2025b) | GameGenX (Che et al., 2025) | GameCraft (Li et al., 2025a) | WorldMem (Xiao et al., 2026) | VMem (Li et al., 2025b) | WorldPlay |
|---|---|---|---|---|---|---|---|
| Resolution | 544p | 360p | 720p | 720p | 360p | 576p | 720p |
| Action Space | Text | Dis. | Dis. | Con. | Dis. | Con. | Con. + Dis. |
| Real-time | ✔ | ✔ | ✗ | ✗ | ✗ | ✗ | ✔ |
| Long-term Consistency | ✗ | ✗ | ✗ | ✗ | ✔ | ✔ | ✔ |
| Long-Horizon | ✗ | ✔ | ✔ | ✔ | ✗ | ✗ | ✔ |
| Domain | General | General | General | General | Minecraft | Static Scene | General |

ited length videos, laying the foundation for world models. With the advancement of powerful architectures (Peebles & Xie, 2023) and sophisticated data pipelines, (Deepmind, 2025; Wan et al., 2025; Kuaishou, 2024; Minimax, 2024; Gao et al., 2025; Kong et al., 2024a), which are trained on web-scale datasets, have demonstrated emergent zero-shot capabilities to perceive, model, and manipulate the visual world (Wiedemer et al., 2025), making it feasible to simulate the physical world.

**Interactive and Consistent World Models.** World models aim to predict future states based on current and past states. Studies such as (Alonso et al., 2024; Bar et al., 2025; Valevski et al., 2025; Yu et al., 2025c; Sun et al., 2025b; He et al., 2025a; Wang et al., 2024b; Miyato et al., 2024; Kong et al., 2024b; Li et al., 2025c; Bahmani et al., 2025; Sun et al., 2025a; Mao et al., 2025b;a; Xiang et al., 2025; Tang et al., 2025) adopt discrete, continuous action signals or text instructions to enable agents to navigate and interact with virtual environments. (Yesiltepe et al., 2026) proposes a training-free framework for instruction-controllable video generation. Subsequent works that aim to achieve geometric consistency can be categorized into two types: explicit 3D reconstruction and implicit conditioning. (Li et al., 2025b; Yu et al., 2025a; Ren et al., 2025; Cao et al., 2025; Yu et al., 2025; YU et al., 2025; Liu et al., 2026a) ensure spatial consistency by explicitly reconstructing 3D representations and rendering condition frames from these representations. However, they heavily rely on reconstruction quality, making it challenging to maintain long-term consistency. Recent work (HunyuanWorld, 2025) constructs 3D world models explicitly, without relying on video generation models. Although achieving promising 3D generation results, they can not be performed in real-time. In contrast, (Xiao et al., 2026; Yu et al., 2025b) achieve implicit conditioning by leveraging field-of-view (FOV) to retrieve relevant context from historical frames. Concurrent work (Hong et al., 2025) achieves interactive generation with fixed-length consistency through context compression (Zhang et al., 2025c). However, developing a real-time world model with long-horizon geometric consistency remains unsolved.

**Distillation.** For video diffusion models, existing ap-

proaches typically employ distillation (Salimans & Ho, 2022; Geng et al., 2025; Frans et al., 2025; Li et al., 2026; Zhang et al., 2025a) to achieve few-step inference, achieving faster generation. (Sauer et al., 2024a;b; Kang et al., 2024; Lin et al., 2025a; 2024; 2025b) adopt adversarial training strategies to enable few-step inference, however, they often suffer from training instability and mode collapse. (Yin et al., 2024b;a; Lu et al., 2025; Shin et al., 2026) utilize Variational Score Distillation (Wang et al., 2023) to achieve outstanding few-step generation performance in various tasks. In addition, CausVid (Yin et al., 2025) proposes distilling a causal student model from a bidirectional teacher diffusion model to achieve real-time autoregressive generation. Furthermore, Self-Forcing (Huang et al., 2025c) mitigates exposure bias by refining the rollout strategy of CausVid. Our method proposes context forcing to preserve both the interactivity and geometric consistency while achieving real-time generation.

## 3. Method

Our goal is to construct a geometry-consistent and real-time interactive world model $N_\theta(x_t|O_{t-1}, A_{t-1}, a_t, c)$ parameterized by $\theta$, which can generate next chunk $x_t$ (a chunk is a few frames) based on past observations $O_{t-1} = \{x_{t-1}, ..., x_0\}$, action sequences $A_{t-1} = \{a_{t-1}, ..., a_0\}$, and current action $a_t$. Here, $c$ is a text prompt or image that describes the world. For simplicity of notation, we omit $A, a, c$ in following sections. We first introduce the relevant preliminaries in Sec. 3.1. In Sec. 3.2, we discuss the action representation for control. Sec. 3.3 describes our reconstituted context memory to ensure long-term geometric consistency, followed by Sec. 3.4 covering our context forcing, which mitigates exposure bias and enables few-step generation while maintaining long-term consistency. Finally, Sec. 3.5 details additional optimizations for real-time streaming generation. The pipeline is shown in Fig. 2.

### 3.1. Preliminaries

**Full-sequence Video Diffusion Model.** Current video diffusion models (Kong et al., 2024a; Wan et al., 2025) typically

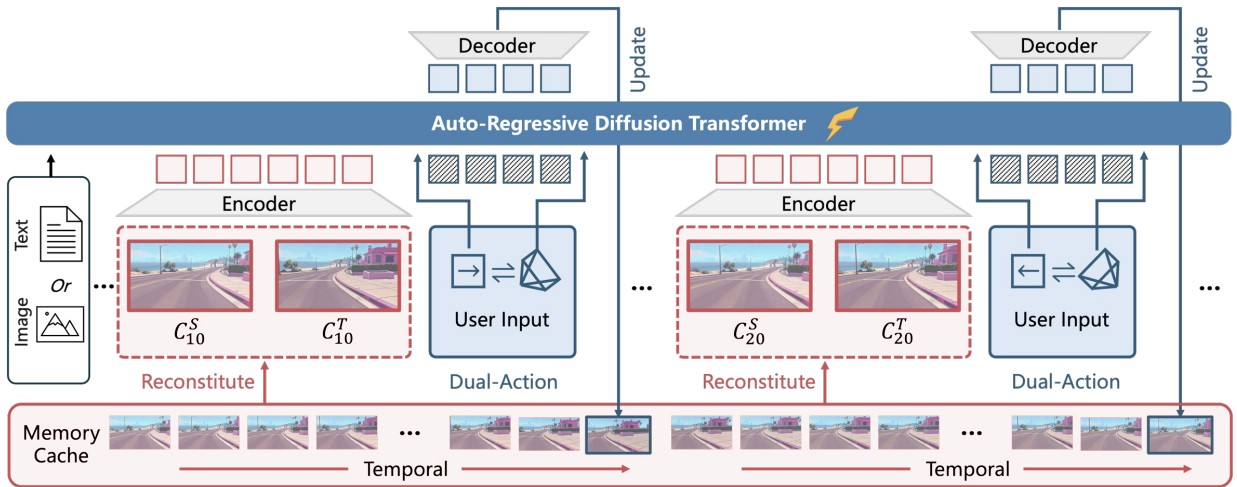

*Figure 2.* **Method overview.** Given a single image or text prompt to describe a world, **WorldPlay** performs a next chunk (16 video frames) prediction task to generate future videos conditioned on action from users. For the generation of each chunk, we dynamically reconstitute context memory from past chunks to enforce long-term temporal and geometric consistency.

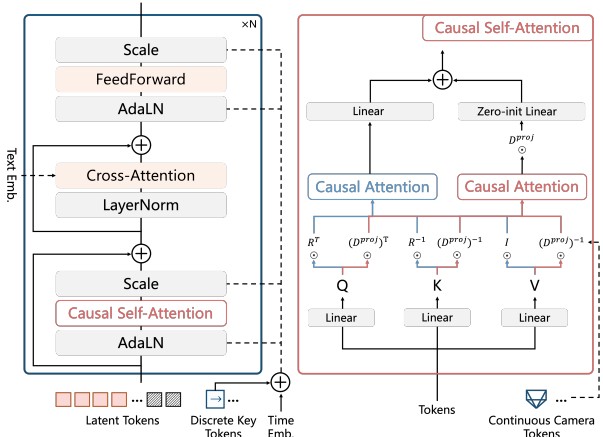

*Figure 3.* Detailed architecture of our autoregressive diffusion transformer. The discrete key is incorporated with time embedding, while the continuous camera pose is injected into causal self-attention through PRoPE (Li et al., 2025c).

consist of a causal 3D VAE (Kingma & Welling, 2013) and a Diffusion Transformer (DiT) (Peebles & Xie, 2023), where each DiT block is composed of 3D self-attention, cross-attention, and feedforward network (FFN). The diffusion timestep is processed by positional embedding (PE) and a Multi-Layer Perceptron (MLP) to modulate the DiT blocks. The model is trained using flow matching (Lipman et al., 2023). Specifically, given a video latent $z_0$ encoded by the 3D VAE, a random noise $z_1 \sim \mathcal{N}(0, I)$, and a diffusion timestep $k \in [0, 1]$, an intermediate latent $z_k$ is obtained through linear interpolation. The model is trained to predict the velocity $v_k = z_0 - z_1$,

$$\mathcal{L}_{\text{FM}}(\theta) = \mathbb{E}_{k,z_0,z_1} \left\| N_\theta(z_k, k) - v_k \right\|^2. \quad (1)$$

**Chunk-wise Autoregressive Generation.** However, the full-sequence video diffusion model is a non-causal ar-

chitecture, which limits its ability for infinite-length interactive generation. Inspired by Diffusion Forcing (Chen et al., 2024a), we finetune it into a chunk-wise autoregressive video generation model. Specifically, for video latent $z_0 \in \mathbb{R}^{C \times T \times H \times W}$, we divide it into $\frac{T}{4}$ chunks $\{z_0^i \in \mathbb{R}^{C \times 4 \times H \times W} | i = 0, ..., \frac{T}{4} - 1\}$, and thus each chunk (4 latents) can be decoded into 16 frames. During training, we add different noise levels $k_i$ for each chunk and modify the full-sequence self-attention to block causal attention. The training loss is similar to Eq. 1.

### 3.2. Dual Action Representation for Control

Existing methods use keyboard and mouse inputs as action signals and inject the action control via MLP (Decart, 2024; Xiao et al., 2026) or attention blocks (He et al., 2025b; Yu et al., 2025b). This enables the model to learn physically plausible movements across scenes with diverse scales (*e.g.* very large and small scenes). However, they struggle to provide precise previous locations for spatial memory retrieval. In contrast, camera poses (rotation matrix and translation vector) provide accurate spatial locations that facilitate precise control and memory retrieval, but training only with camera poses faces challenges in training stability due to the scale variance in the training data. To address this, we propose a dual action representation that combines the best of both worlds as shown in Fig. 3. This design not only caches spatial locations for our memory module in Sec. 3.3, but also enables robust and precise control. Specifically, we employ PE and a zero-initialized MLP to encode discrete keys and incorporate it into the timestep embedding, which is then used to modulate the DiT blocks. For continuous camera pose, we leverage relative positional encoding, *i.e.*, PRoPE (Li et al., 2025c), which offers greater generalizability than commonly used raymaps, to inject complete

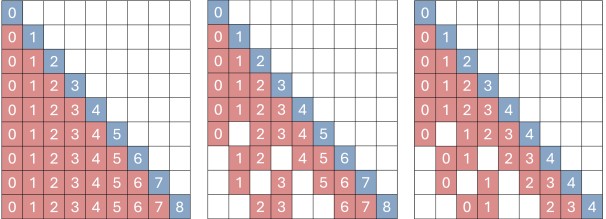

*(a)* Full context   *(b)* Absolute indices   *(c)* Relative indices

*Figure 4.* **Memory mechanism comparisons**. The red and blue blocks represent the memory and current chunk, respectively. The number in each block represents the temporal index in RoPE. For simplicity of illustration, each chunk only contains one frame.

camera frustums into self-attention blocks. The original self-attention computation is as follows,

$$Attn_1 = Attn(R^\top \odot Q, R^{-1} \odot K, V), \qquad (2)$$

where $R$ represents the 3D rotary PE (RoPE) (Su et al., 2024) for video latents. To encode frustum relationships between cameras, we utilize an additional attention,

$$\begin{aligned} Attn_2 = &D^{proj} \odot Attn((D^{proj})^\top \odot Q, \\ &(D^{proj})^{-1} \odot K, (D^{proj})^{-1} \odot V), \end{aligned} \qquad (3)$$

here, $D^{proj} = \begin{bmatrix} K & \mathbf{0} \\ \mathbf{0} & 1 \end{bmatrix} T^{cw}$ is derived from the camera's intrinsic $K$ and extrinsic parameters $T^{cw}$, as described in (Li et al., 2025c). The result of each self-attention block is $Attn_1 + zero\_init(Attn_2)$.

### 3.3. Reconstituted Context Memory for Consistency

Maintaining long-term geometric consistency requires recalling past frames, ensuring content remains unchanged when revisiting to a previous location. However, naively using all past frames as context (Fig. 4a) is computationally intractable and redundant for long sequences. To address this, we rebuild a memory context $C_t$ from past chunks $O_{t-1}$ for each new chunk $x_t$. Our approach advances beyond prior work (Xiao et al., 2026; Yu et al., 2025b; Chen et al., 2025) by combining both short-term temporal cues and long-range spatial references: 1) A temporal memory $(C_t^T)$ comprises $L$ most recent chunks $\{x_{t-L}, ..., x_{t-1}\}$ to ensure short-term motion smoothness. 2) A spatial memory $(C_t^S)$ samples from non-adjacent past frames to prevent geometric drift over long sequences, where $C_t^S \subseteq O_{t-1} - C_t^T$. This sampling is guided by geometric relevance scores that incorporate both FOV overlap and camera distance.

Once memory context is rebuilt, the challenge shifts to applying them to enforce consistency. Effectively using retrieved context requires overcoming a fundamental flaw in positional encodings. With standard RoPE (Fig.4b), the distance between the current chunk and past memory grows

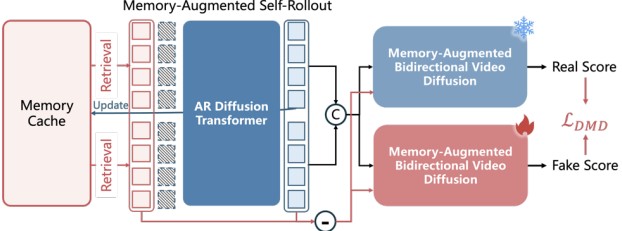

*Figure 5.* **Context forcing** is a novel distillation method that employs memory-augmented self-rollout and memory-augmented bidirectional video diffusion to preserve long-term consistency, enable real-time interaction, and mitigate error accumulation.

unbounded over time. This growing relative distance can eventually exceed the trained interpolation range in RoPE, causing extrapolation artifacts (Su et al., 2024). More critically, the growing perceived distance to these long-past spatial memory would weaken their influence on the current prediction. To resolve this, we propose Temporal Reframing (Fig.4c). We discard the absolute temporal indices, and dynamically re-assign new positional encodings to all context frames, establishing a fixed, small relative distance to the current, irrespective of their actual temporal gap. This operation effectively "pulls" important past frames closer in time, ensuring their sustained influence and enabling robust extrapolation for long-term consistency.

### 3.4. Context Forcing

Autoregressive models often suffer from error accumulation during long video generation, leading to degraded visual quality over time (Huang et al., 2025c; Yin et al., 2025). Moreover, the multi-step denoising of diffusion models is too slow for real-time interaction. Recent methods (Huang et al., 2025c; Yang et al., 2026; Liu et al., 2026b; Cui et al., 2026a) address these challenges by distilling a powerful bidirectional teacher diffusion model into a fast, few-step autoregressive student. These techniques force the student's output distribution $p_\theta(x_{0:t})$ to align with the teacher's, thereby improving generation quality by employing a distribution matching loss (Yin et al., 2024a):

$$\nabla_\theta \mathcal{L}_{DMD} = \mathbb{E}_k(\nabla_\theta \mathrm{KL}(p_\theta(x_{0:t})||p_{data}(x_{0:t}))), \qquad (4)$$

where the gradient of the reverse KL can be approximated by the score difference derived from teacher model.

However, these methods are incompatible with memory-aware models due to a critical distribution mismatch. Standard teacher diffusion models are trained on short clips and are inherently memory-less. Even if a teacher is augmented with memory, its bidirectional nature inevitably differs from the student's causal, autoregressive process. This means that without a meticulously designed memory context to mitigate this gap, the difference in memory context will make their conditional distributions $p(x|C)$ misaligned, which in turn causes distribution matching to fail.

*Table 2.* **Quantitative comparisons.** We compare against both methods without memory, *i.e.*, CameraCtrl (He et al., 2025a), SEVA (Zhou et al., 2025), ViewCrafter (Yu et al., 2025), Matrix-Game-2.0 (He et al., 2025b), and GameCraft (Li et al., 2025a), and methods with memory, *i.e.*, Gen3C (Ren et al., 2025), VMem (Li et al., 2025b). Our method achieves superior results, particularly in long-term settings, which more clearly demonstrate the long-term consistency.

| | Real-time | Short-term (61 frames) | | | | | Long-term ($\geq$ 250 frames) | | | | |
|---|---|---|---|---|---|---|---|---|---|---|---|
| | | PSNR ↑ | SSIM ↑ | LPIPS ↓ | $R_{\text{dist}}$ ↓ | $T_{\text{dist}}$ ↓ | PSNR ↑ | SSIM ↑ | LPIPS ↓ | $R_{\text{dist}}$ ↓ | $T_{\text{dist}}$ ↓ |
| CameraCtrl (He et al., 2025a) | ✗ | 17.93 | 0.569 | 0.298 | 0.037 | 0.341 | 10.09 | 0.241 | 0.549 | 0.733 | 1.117 |
| SEVA (Zhou et al., 2025) | ✗ | 19.84 | 0.598 | 0.313 | 0.047 | 0.223 | 10.51 | 0.301 | 0.517 | 0.721 | 1.893 |
| ViewCrafter (Yu et al., 2025) | ✗ | 19.91 | 0.617 | 0.327 | 0.029 | 0.543 | 9.32 | 0.277 | 0.661 | 1.573 | 3.051 |
| Gen3C (Ren et al., 2025) | ✗ | 21.68 | 0.635 | 0.278 | **0.024** | 0.477 | 15.37 | 0.431 | 0.483 | 0.357 | 0.979 |
| VMem (Wang et al., 2024b) | ✔ | 19.97 | 0.587 | 0.316 | 0.048 | 0.219 | 12.77 | 0.335 | 0.542 | 0.748 | 1.547 |
| Matrix-Game-2.0 (He et al., 2025b) | ✔ | 17.26 | 0.505 | 0.383 | 0.287 | 0.843 | 9.57 | 0.205 | 0.631 | 2.125 | 2.742 |
| GameCraft (Li et al., 2025a) | ✗ | 21.05 | 0.639 | 0.341 | 0.151 | 0.617 | 10.09 | 0.287 | 0.614 | 2.497 | 3.291 |
| Ours (w/o Context Forcing) | ✗ | 21.27 | 0.669 | 0.261 | 0.033 | 0.157 | 16.27 | 0.425 | 0.495 | 0.611 | 0.991 |
| Ours (full) | ✔ | **21.92** | **0.702** | **0.247** | 0.031 | **0.121** | **18.94** | **0.585** | **0.371** | **0.332** | **0.797** |

We thus propose context forcing as shown in Fig. 5, which alleviates the memory context misalignment between teacher and student for distillation. For the student model, we self-rollout 4 chunks conditioned on the memory context $p_\theta(x_{j:j+3}|x_{0:j-1}) = \prod_{i=j}^{j+3} p_\theta(x_i|C_i)$. To construct our teacher model $V_\beta$, we augment a standard bidirectional diffusion model with memory, and structure its context by masking $x_{j:j+3}$ from student's memory context,

$$p_{data}(x_{j:j+3}|x_{0:j-1}) = p_\beta(x_{j:j+3}|C_{j:j+3} - x_{j:j+3}), \quad (5)$$

where $C_{j:j+3}$ denotes all context memory chunks corresponding to student's self-rollout $x_{j:j+3}$. By aligning the memory context with the student model, we enforce the distributions represented by the teacher to be as close as possible to the student model, which enables more effectively distribution matching. Moreover, this avoids training $V_\beta$ on long videos and redundant context, facilitating the learning of long-term visual distribution. Additionally, we introduce a progressive distillation strategy that incrementally increases the number of self-rollout latents. This facilitates distillation across varying sequence lengths, thereby enhancing long-horizon video generation. Through context forcing, we preserve long-term consistency in real-time generation with 4-denoising steps, and mitigate error accumulation.

### 3.5. Streaming Generation with Real-Time Latency

We augment context forcing with a suite of optimizations to minimize latency, unlocking an interactive streaming experience at 24 FPS and 720p resolution on 8×H800 GPUs.

**Mixed Parallelism Method for DiT and VAE.** Unlike the conventional parallelism method that replicates the entire model or adapting sequence parallelism on the temporal dimension, our parallelism method combines sequence parallelism (Li et al., 2023) and attention parallelism, which partitions the tokens of each chunk across devices. This design ensures that the computational workload is distributed evenly, substantially reducing per-chunk inference time

while maintaining generation quality.

**Streaming Deployment and Progressive Decoding.** To minimize time-to-first-frame and enable seamless interaction, we adopt a streaming deployment architecture using NVIDIA Triton Inference Framework and implement a progressive VAE decoding strategy that decodes and streams frames in smaller batches, allowing users to observe generated content while subsequent frames are still being processed. This streaming pipeline ensures smooth, low-latency interaction even under varying computational loads.

**Quantization and Efficient Attention.** Furthermore, we employ a comprehensive suite of quantization strategies. Specifically, we adopt Sage Attention (Zhang et al., 2025b), float quantization, and matrix multiplication quantization to improve the inference performance. Additionally, we use KV-cache mechanisms for attention modules to eliminate redundant computations during autoregressive generation.

## 4. Experiments

**Implementation Details.** WorldPlay is trained on a comprehensive dataset comprising approximately 320K high-quality video samples derived from both real-world footage and synthetic environments. Details regarding the dataset processing pipeline and the training/inference are provided in Appendix B and Appendix A, respectively.

**Evaluation Protocol.** Our test set comprises 600 cases sourced from DL3DV, game videos, and AI-generated images spanning a range of styles. For the short-term setting, we utilize the camera trajectories from the test videos as the input pose. The generated frames are directly compared against the ground truth to assess visual quality and action precision. For the long-term setting, we test the long-horizon generation ability and long-term consistency using various custom cycle camera trajectories designed to enforce revisiting. Each model generates frames along a customize trajectory and then returns along the same path, metrics are evaluated on the return path by comparing the generated

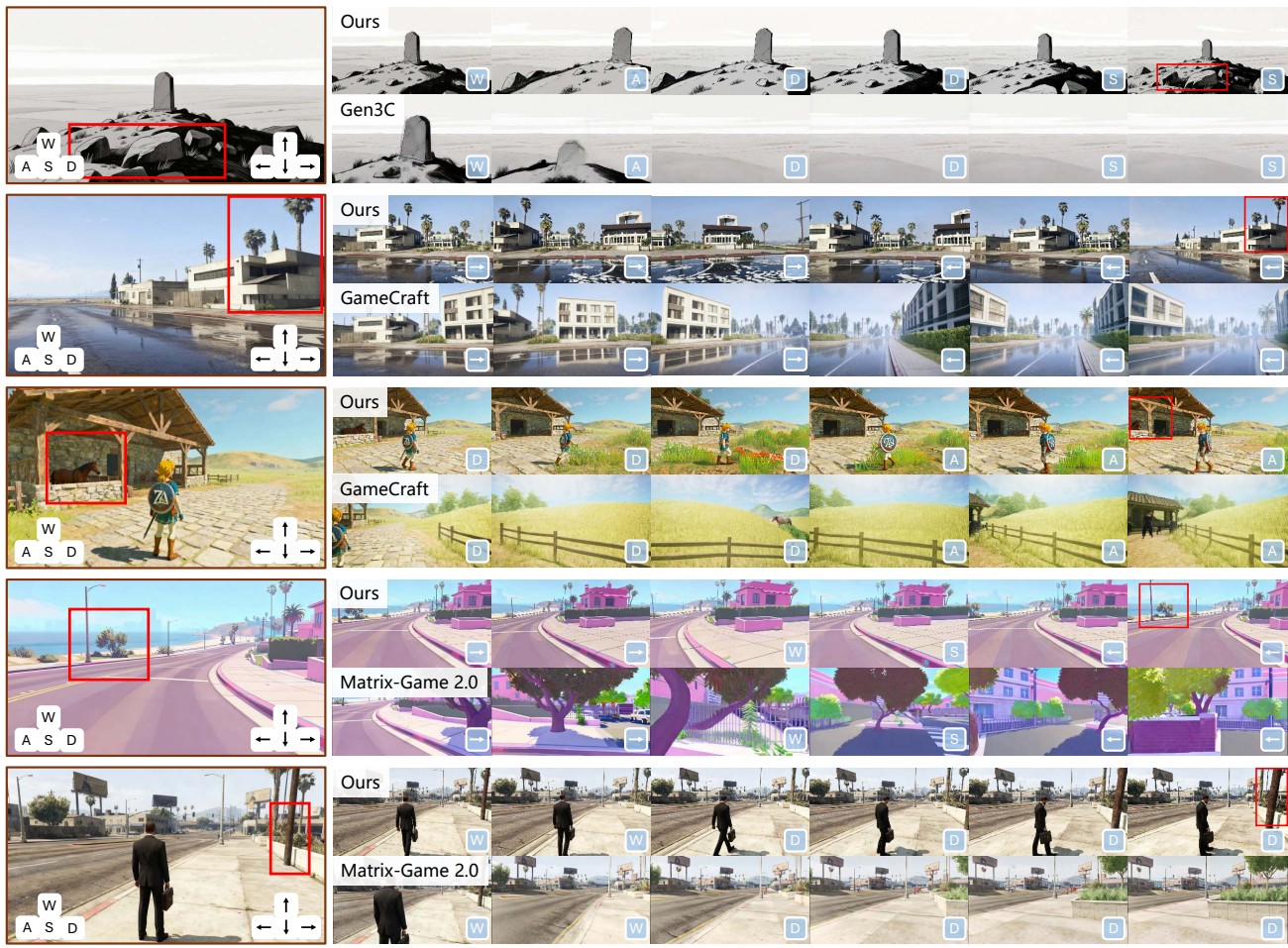

*Figure 6.* **Qualitative comparisons with existing methods.** WorldPlay achieves the state-of-the-art long-term consistency (*shown in red boxes*) and visual quality across diverse scenes, including both first- and third-person real and stylized worlds.

frame to the corresponding frame generated during the initial pass. We employ LPIPS, PSNR, and SSIM to measure visual quality and $R_{\text{dist}}$ and $T_{\text{dist}}$ to quantify action precision.

Specifically, we use the ground truth poses as the input for each model to generate videos. Then, we utilize ViPE to estimate the camera pose of the generated videos. Following the previous works (He et al., 2025a; Yu et al., 2025), we compute the relative poses of the ground truth and generated camera poses by setting the extrinsic matrix of first frame as an identity matrix and normalize the translation scale using the furthest frame. The rotation distance $R_{\text{dis}}$ is calculated by comparing the predicted rotation matrices $R_{gen}$ and ground truth rotation matrices $R_{gt}$:

$$R_{\text{dis}} = \arccos\left(\frac{\text{tr}(\mathbf{R}_{\text{gen}}\mathbf{R}_{\text{gt}}^{\top}) - 1}{2}\right), \qquad (6)$$

where $\text{tr}(\cdot)$ denotes the trace of the matrix. The translation distance $T_{dis}$ is computed between the predicted $\mathbf{t}_{\text{gen}}$ and

ground truth translation vectors $\mathbf{t}_{\text{gt}}$:

$$T_{\text{dis}} = \|\mathbf{t}_{\text{gt}} - \mathbf{t}_{\text{gen}}\|_2^2. \qquad (7)$$

**Baselines.** We conduct comprehensive comparisons against various baselines, which mainly fall into two categories: 1) *Action-controlled diffusion models without memory*: CameraCtrl (He et al., 2025a), SEVA (Zhou et al., 2025), ViewCrafter (Yu et al., 2025), Matrix-Game 2.0 (He et al., 2025b) and GameCraft (Li et al., 2025a); 2) *Action-controlled diffusion models with memory*: Gen3C (Ren et al., 2025) and VMem (Li et al., 2025b). More evaluation results can be found in Appendix.

### 4.1. Main Result

**Quantitative Results.** As shown in Table 2, in the short-term regime, our approach achieves superior visual fidelity and maintains competitive control accuracy. Although methods leveraging explicit 3D representations (*i.e.*ViewCrafter (Yu et al., 2025), Gen3C (Ren et al., 2025)) realize more accurate rotation, they suffer from issues such

*Table 3.* **Quantitative Comparison of 3D Structural Consistency** using MEt3R (Asim et al., 2025).

| Method | MEt3R Score ($\downarrow$) |
|---|---|
| Gen3C (Ren et al., 2025) | 0.187 |
| Matrix-Game-2.0 (He et al., 2025b) | 0.367 |
| GameCraft (Li et al., 2025a) | 0.305 |
| **Ours (full)** | **0.133** |

*Table 4.* **Quantitative evaluation of inference acceleration strategies.** "P" and "Q" denote Parallelism and Quantization, respectively.

| DiT P&Q | VAE P&Q | Streaming Decoding | FPS | Improvement |
|---|---|---|---|---|
| | | | 2.80 | — |
| ✓ | | | 3.80 | $1.35\times$ |
| ✓ | ✓ | | 16.0 | $5.71\times$ |
| ✓ | ✓ | ✓ | 24.0 | $8.57\times$ |

as the inaccurate depth estimation and inconsistent scale when performing translations. For more challenging long-term scenarios, where action accuracy generally degrades, our method remains more stable and achieves the best performance. Regarding long-term geometric consistency, Matrix-Game-2.0 (He et al., 2025b) and GameCraft (Li et al., 2025a) exhibit poor performance due to the lack of memory mechanism. Although VMem (Li et al., 2025b) and Gen3C (Ren et al., 2025) employ explicit 3D cache to maintain consistency, they are constrained by depth accuracy and alignment, making it difficult to achieve robust long-term consistency. Benefiting from Reconstituted Context Memory, we achieve improved long-term consistency. Moreover, through context forcing, we further prevent error accumulation, resulting in better visual quality and action accuracy. To rigorously evaluate the long-horizon 3D structural consistency of our model, we incorporate a more advanced metric, MEt3R (Asim et al., 2025), which explicitly models the multi-view correspondence by leveraging pre-trained 3D geometric priors (e.g., DUSt3R (Wang et al., 2024a)). We evaluate the generated long-horizon videos by pairing frames from the initial trajectory with the corresponding frames in the revisiting trajectory, which are then fed into the MEt3R model to quantify their 3D structural alignment. As shown in Table. 3, it clearly demonstrates the superiority of our approach in maintaining precise 3D consistency.

To further validate the efficiency of our optimization strategies proposed in Sec. 3.5, we conduct experiments on inference speed as detailed in Table 4, demonstrating that the tailored parallelization and quantization applied to DiT and VAE significantly boost the inference throughput of our model. Crucially, WorldPlay concurrently achieves the requisite real-time interactivity for immersive simulation.

**Qualitative Results.** We provide qualitative comparisons with baselines in Fig. 6. The explicit 3D cache used in Gen3C (Ren et al., 2025) is highly sensitive to the quality of intermediate output and limited by the accuracy of depth

*Table 5.* **Ablation for action representation.**

| Action | PSNR$\uparrow$ | SSIM$\uparrow$ | LPIPS$\downarrow$ | $R_{\mathrm{dist}} \downarrow$ | $T_{\mathrm{dist}} \downarrow$ |
|---|---|---|---|---|---|
| Discrete | 21.47 | 0.661 | 0.248 | 0.103 | 0.615 |
| Continuous | 21.93 | 0.665 | 0.231 | 0.038 | 0.287 |
| Full | **22.09** | **0.687** | **0.219** | **0.028** | **0.113** |

*Table 6.* **Ablation for positional encoding design in memory.** The results are evaluated on the long-term test data.

| | PSNR$\uparrow$ | SSIM$\uparrow$ | LPIPS$\downarrow$ | $R_{\mathrm{dist}} \downarrow$ | $T_{\mathrm{dist}} \downarrow$ |
|---|---|---|---|---|---|
| RoPE | 14.03 | 0.358 | 0.534 | 0.805 | 1.341 |
| Reframed RoPE | **16.27** | **0.425** | **0.495** | **0.611** | **0.991** |

estimation. Conversely, our reconstituted context memory guarantees long-term consistency with more robust implicit prior, achieving superior generalizability. Matrix-Game-2.0 (He et al., 2025b) and GameCraft (Li et al., 2025a) fail to support free exploration due to the lack of memory. Furthermore, they do not generalize well to third-person scenarios, making it difficult to control agents and limiting their applicability. In contrast, WorldPlay successfully extends its efficacy to these scenarios and maintains high visual fidelity and long-term geometric consistency.

### 4.2. Ablation

**Action Representation.** Table 5 validates the effectiveness of the proposed dual-action representation. When using only discrete keys as action signals, the model struggles to achieve fine-grained control, such as the distance of movement or the degree of rotation, resulting in poor performance on $R_{\mathrm{dist}}$ and $T_{\mathrm{dist}}$ metrics. Using continuous camera poses yields better results but converges more difficult due to scale variance. By employing the dual-action representation, we achieve the best overall control performance.

**RoPE Design.** Table. 6 presents the quantitative results of different RoPE designs within the memory mechanism as detailed in Sec. 3.3, showing that reframed rope outperforms naive counterparts, especially on visual metrics. As illustrated in the upper part of Fig. 7, RoPE is more prone to error accumulation. It also increases the distance between memory and predicted chunk due to absolute temporal indices, resulting in weaker geometric consistency, as shown in the lower part of Fig. 7.

**Context Forcing.** To verify the importance of memory alignment, we train the teacher model following (Yu et al., 2025b), where the memory is selected at latent level rather than at chunk level. Although this may reduce the number of memory context in the teacher model, it also introduce misaligned context between the teacher and student model, leading to collapsed results as shown in Fig. 8a. Moreover, utilizing a memory-less bidirectional model as the teacher induces a distribution mismatch, which hinders long-horizon video generation and significantly compromises long-term consistency, as illustrated in Fig. 8b. Additionally, for the

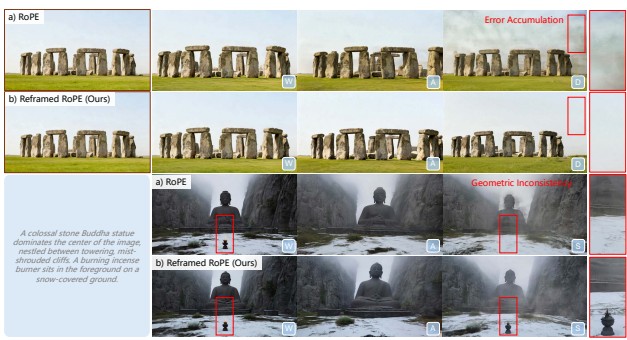

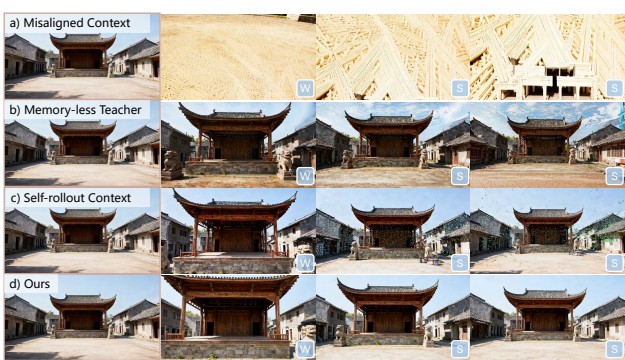

Figure 7. **RoPE design comparisons. Upper:** Our reframed RoPE avoids exceeding the the positional range in standard RoPE, alleviating error accumulation. **Bottom:** By maintaining a small relative distance to long-range spatial memory, it achieves better long-term consistency.

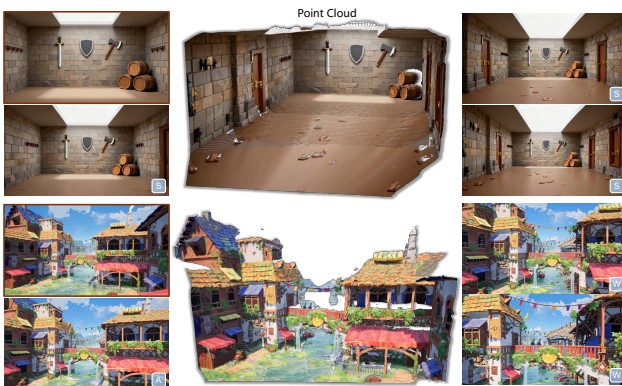

Figure 9. **3D reconstruction results.** We first utilize our model to autoregressively generate videos. The videos are then processed by a 3D reconstruction model to produce the final point clouds.

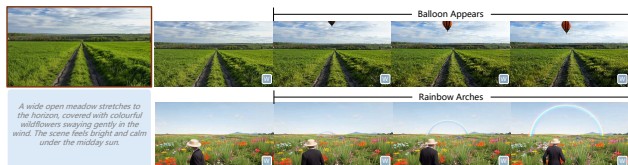

Figure 10. **Promptable event.** Our method supports text-based manipulation during streaming.

at any time to responsively alter the ongoing stream.

## 5. Conclusion

WorldPlay is a powerful world model with real-time interaction and long-term geometric consistency. It empowers users to customize unique worlds from a single image or text prompt. While focused on navigation control, its architecture has shown potential for richer interaction like dynamic, text-triggered events. By providing a systematic framework for control, memory, and distillation, WorldPlay marks a critical step toward creating consistent and interactive virtual worlds.

**Limitations.** While WorldPlay demonstrates strong performance, several avenues remain open for exploration and improvement. First, our model can generawting videos of approximately 30 seconds, efficiently scaling this framework to longer durations, such as minutes or hours (Cui et al., 2026a;b), remains a significant challenge. Second, although distillation is utilized to mitigate error accumulation, fundamentally averting this phenomenon during the training of autoregressive diffusion models remains a critical open challenge. Moreover, expanding the action types to a broader set with multi-agent interaction and complex physical dynamics is another promising direction. Finally, retrieval mechanisms based on the FOV may fail to accurately identify memory context when faced with significant occlusions (Yu et al., 2025b).

---

Figure 8. **Ablation for context forcing. a)** When the teacher and student have misaligned context, it leads to distillation failure, resulting in collapsed outputs. **b)** Leveraging memory-less teacher introduces a distribution mismatching. **c)** Self-rollout historical context can introduce artifacts. Zoom in for details.

past chunks $x_{0:j-1}$, we attempt to self-rollout historical chunks as context following (Yang et al., 2026). However, this may cause the bidirectional diffusion model to provide inaccurate score estimation, as it is trained using clean chunks as memory. Consequently, this discrepancy introduces artifacts as illustrated in Fig. 8c. We obtain historical chunks by sampling from real videos, which yields superior results as shown in Fig. 8d.

### 4.3. Application

**3D Reconstruction.** Benefiting from the long-term geometric consistency, we can integrate a feed-forward 3D reconstruction model (Liu et al., 2025) to produce high-quality point clouds from the generated videos, as presented in Fig. 1 (d) and Fig. 9.

**Promptable Event.** Beyond navigation control, WorldPlay supports text-based interaction to trigger dynamic world events (*i.e.*, environmental transitions and object appearances). As shown in Fig. 10 and Fig. 1 (e), users can prompt

## Acknowledgement

This work was supported by the Hong Kong Research Grants Council under the Areas of Excellence scheme grant AoE/E-601/22-R and NSFC/RGC Collaborative Research Scheme grant CRS_HKUST603/22.

## Impact Statement

This paper presents work whose goal is to advance the field of Machine Learning. There are many potential societal consequences of our work, none which we feel must be specifically highlighted here.

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

# A. Training and Inference Details

We adopt the pretrained DiT-based video diffusion models (Wan et al., 2025; Kong et al., 2024a) as the backbone. For the chunk-wise autoregressive diffusion transformer, we group 4 latents into a chunk. For the memory context, we set the temporal memory length to 3 chunks and the spatial memory length to 1 chunk. Moreover, inspired by (Yang et al., 2026; Liu et al., 2026b; Xiao et al., 2024), we observe that preserving the first chunk as attention sink enhances long-term consistency and further mitigates error accumulation. For the bidirectional teacher model $V_\beta$, we also adopt the dual-action representation and construct the memory context as described in Sec. 3.4. The training consists of three stages.

**Stage One: Action Control.** In the first stage, we focus on injecting action control into the pretrained model. We employ the dual action representation to the pretrained model and train the bidirectional action model for 30K iterations. Then, we replace the 3D self-attention with block causal attention and train for an additional 30K iterations as our AR action model. We find that this enables the AR action model to converge more easily. In this stage, the model is trained on 61 frames (4 chunks) using the Adam optimizer with a learning rate of $1e-5$ and a batch size of 64.

**Stage Two: Memory.** In the second stage, we train the bidirectional action model and the AR action model with context memory as described in Sec.3.3 and Sec.3.4, respectively. For the bidirectional action model, the generation sequence consists of 4 chunks $x_{j:j+3}$ (16 latents, 61 frames). We utilize $C_{j:j+3} - x_{j:j+3}$ as a variable-length memory context for the teacher. Following the Flow Matching framework, we apply noise from $[0, 1]$ to $x_{j:j+3}$, while the memory context $C_{j:j+3} - x_{j:j+3}$ undergoes uniform noise sampling from $[0, 0.2]$ to enable robust conditioning. Crucially, the training loss are computed strictly on the generation sequence. In stage two, both the bidirectional and AR models are trained on sequences of up to 160 latents (637 frames). Other settings remain the same as in the first stage.

**Stage Three: Context Forcing.** In the final stage, we use the bidirectional model as the teacher and the AR model as the student for distillation. To stabilize the distillation process, we employ a progressive training strategy that gradually increases the maximum length of the generated latents. For the student model, the learning rate is set to $1e-6$, while for the bidirectional model, which is used to compute the fake score, the learning rate is set to $2e-7$. The models are trained for 2K iteration with a batch size of 64. All other hyperparameters follow (Huang et al., 2025c). For the details of context forcing, see Algorithm 1.

Finally, our AR model can produce multiple chunks in a streaming fashion with KV cache as shown in Algorithm 2. When the user provides only camera poses, we first compute the relative translations and rotations between consecutive poses, and then apply a thresholding mechanism to identify and convert them into discrete actions. Conversely, when only discrete actions are available, we use the predefined relative translations and rotations associated with each action to convert them into camera poses.

---

**Algorithm 1** Context Forcing Training

**Require:** Number of denoising timesteps $d$ and chunks $n = 4$
**Require:** Dataset $D$ (encoded by 3D VAE)
**Require:** AR diffusion model $N_\theta$
**Require:** Bidirectional diffusion model $V_\beta^{fake}$ and $V^{real}$
1:  **loop**
2:      Progressively increase maximum chunk length $m$
3:      Sample chunk length $j \sim \text{Uniform}(0, 1, \ldots, m)$
4:      Sample context $x_{0:j-1} \sim D$
5:      **for** $i = j, \ldots, j + n - 1$ **do**
6:          Initialize $x_i^{init} \sim \mathcal{N}(0, I)$
7:          Reconstitute context memory $C_i \subseteq \{x_0, \ldots, x_{i-1}\}$
8:          Sample $s \sim \text{Uniform}(1, 2, \ldots, d)$
9:          Self-rollout $x_i$ using $N_\theta$ with $C_i$ and $s$ denoising steps
10:     **end for**
11:     Align context memory $C^{tea} \leftarrow C_{j:j+n-1} - x_{j:j+n-1}$
12:     Sample diffusion timestep $k \sim [0, 1]$
13:     $\hat{x}_{j:j+n-1} \leftarrow AddNoise(x_{j:j+n-1}, k)$
14:     Fake score $S^{fake} \leftarrow V_\beta^{fake}(\hat{x}_{j:j+n-1}, C^{tea}, k)$
15:     Real score $S^{fake} \leftarrow V^{real}(\hat{x}_{j:j+n-1}, C^{tea}, k)$
16:     Update $\theta$ via distribution matching loss
17:     Update $\beta$ via flow matching loss (Huang et al., 2025c)
18: **end loop**

---

**Algorithm 2** Inference with KV Cache

**Require:** Number of inference chunks $n_c$
**Require:** Denoise timesteps $\{k_1, \ldots, k_d\}$
**Require:** Number of inference chunks $n_c$
**Require:** AR diffusion model $N_\theta$ (returns KV embeddings via $N_\theta^{KV}$)
1:  Initialize model output $X_\theta \leftarrow []$
2:  Initialize KV cache $\mathbf{KV} \leftarrow []$
3:  **for** $i = 0, \ldots, n_c - 1$ **do**
4:      Initialize $x_i \sim \mathcal{N}(0, I)$
5:      Reconstitute context memory $C_i \subseteq \{x_0, \ldots, x_{i-1}\}$
6:      **for** $s = d, \ldots, 1$ **do**
7:          **if** $s = d$ and $i > 1$ **then**
8:              Reset $\mathbf{KV} \leftarrow N_\theta^{KV}(C_i, 0)$
9:          **end if**
10:         Denoise $x_i \leftarrow N_\theta(x_i, \mathbf{KV}, k_s)$
11:     **end for**
12:     Add output $X_\theta$.append($x_i$)
13: **end for**
14: **return** $X_\theta$

---

*Table 7.* **Data organization.** The table details the four categories of data, their sources, the availability of action annotations (discrete and continuous), the number of clips, and their corresponding ratio in the final dataset.

| Category | Data Source | Annotation (discrete, continuous) | # Clips | Ratios |
|---|---|---|---|---|
| Real-World Dynamics | Sekai (Li et al., 2025d) | (✗, ✗) | 40K | 12.5% |
| Real-World 3D Scene | DL3DV (Ling et al., 2024) | (✔, ✔) | 60K | 18.75% |
| Synthetic 3D Scene | UE Rendering | (✗, ✔) | 50K | 15.625% |
| Simulation Dynamics | Game Video Recordings | (✔, ✗) | 170K | 53.125% |

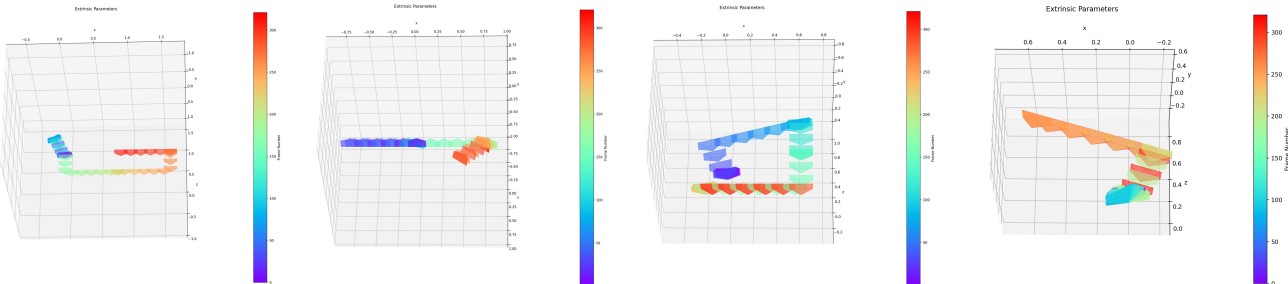

*Figure 11.* **Camera trajectories included in our collected dataset.**

## B. Dataset

Table 7 provides a comprehensive breakdown of our dataset. We deliberately curate a diverse and high-quality collection, encompassing data from the simulation engine and real world, as well as static and dynamic environments, to guarantee the strong generalization of our model.

For Real-World Dynamics, we employ the Sekai dataset (Li et al., 2025d). However, the original videos often suffer from scene clutter and high dynamics. To address these issues, we implement a rigorous filtering pipeline. Specifically, we apply a state-of-the-art object detection model (YOLO (Redmon et al., 2016)) to identify the presence of crowds and vehicles. By setting an empirical threshold, we filter out clips with high densities of moving objects, thereby ensuring annotation accuracy and stable training.

Regarding the Real-World 3D Scene data (DL3DV (Ling et al., 2024)), the original videos lack diversity in camera movement speed and trajectory complexity. To overcome this, we implement a sophisticated processing workflow: 3D Scene Reconstruction → Customized Trajectory Rendering → Visual Quality Filtering → Video Repair Post-processing (using Difix3D+ (Wu et al., 2025)). This procedure yields additional 60K high-quality real video clips featuring balanced movement speed. During the customized trajectory rendering stage, we deliberately design diverse revisit trajectories to facilitate the learning of long-term geometric consistency. The discrete actions and continuous camera poses in these rendered data are highly accurate, which helps the model learn well-structured action patterns.

For Synthetic 3D Scene (UE Rendering) data, we collect hundreds of UE scenes and obtain 50K video clips by rendering complex, customized trajectories. For Simulation Dynamics (Game Video Recordings), we establish a dedicated game recording platform and invite players to record 170K video clips from 1st/3rd-person AAA games.

We segment the original long videos into 30 to 40 seconds clips and employ a vision-language model to produce descriptive text annotations for every clip. Subsequently, we leverage VIPE (Huang et al., 2025a) to generate high-quality camera poses for clips without camera annotations. However, given the long duration and high scene diversity of our dataset, we observe that pose estimation could be inaccurate, *i.e.*, pose collapse. Therefore, we filter out videos whose adjacent frames exhibit erratic camera positions or rotation angles. Specifically, we utilize the Peak-to-Median Ratio (PMR, the ratio of the maximum inter-frame motion to the median) of inter-frame motions as a detection metric. We subsample the predicted poses every four frames and compute the relative transformations to approximate the camera's instantaneous velocity. By evaluating the PMR, we can robustly identify impulsive pose jumps. Clips exhibiting a PMR above a conservative threshold (set to 5.0 in our experiments) are classified as unacceptable and are discarded. Finally, for clips lacking discrete action annotations, we derive them from the continuous camera poses: we project the rotation and translation components onto the $x, y, z$ axes and apply a threshold to map these continuous values into corresponding discrete action states.

*Table 8.* **Left: comparison of Models under Context Forcing.** The results are evaluated on the long-term test data. Student (AR) denotes the AR model before distillation, Teacher (bidirectional) refers to the memory-augmented bidirectional video diffusion model, and Final (distilled) represents the AR model after distillation. NFE denotes the number of function evaluations. **Right: ablation for memory size.** Spa. and Tem. denote the number of chunks in spatial memory and temporal memory, respectively.

| | NFE | PSNR↑ | SSIM↑ | LPIPS↓ | $R_{\text{dist}}$ ↓ | $T_{\text{dist}}$ ↓ |
|---|---|---|---|---|---|---|
| Student (AR) | 100 | 16.27 | 0.425 | 0.495 | 0.611 | 0.991 |
| Teacher (Bidirectional) | 100 | **19.31** | **0.599** | 0.383 | **0.209** | **0.717** |
| Final (Distilled) | 4 | 18.94 | 0.585 | **0.371** | 0.332 | 0.797 |

| Spa. | Tem. | PSNR↑ | SSIM↑ | LPIPS↓ | $R_{\text{dist}}$ ↓ | $T_{\text{dist}}$ ↓ |
|---|---|---|---|---|---|---|
| 3 | 1 | **16.41** | 0.418 | 0.502 | 0.634 | 1.054 |
| 1 | 3 | 16.27 | **0.425** | **0.495** | **0.611** | **0.991** |

Fig. 11 illustrates the camera trajectories. Our dataset contains complex and diverse trajectories, including a large number of revisit trajectories, which enables our model to learn precise action control and long-term geometric consistency.

## C. Additional Experimental Results

### C.1. More Qualitative Results

Fig. 12 illustrates the results of WorldPlay under various actions and virtual environments. As shown in the first three rows, we can interact with complex composite actions, *e.g.*, various combinations of movements. Moreover, WorldPlay can follow intricate trajectories, such as complex rotations and alternating sequences of rotations and movements as demonstrated in the middle six rows. This enhanced control capability is enabled by our dual action representation, which allows for more precise and reliable action guidance. Furthermore, WorldPlay exhibits strong generalization, enabling it to control different types of agents, *e.g.*, human or animals, to roam within the scenes as shown in the last two rows in Fig. 12 and the last two cases in Fig. 13. For more intuitive perspectives, please refer to the supplementary videos.

### C.2. Long Video Generation

Fig. 14 presents long video generation results from World-Play, we maintain long-term consistency, *e.g.*, frame 1 and frame 252 in the top two examples, and preserve high visual quality throughout the entire sequence. Moreover, our context memory ensures that the generation time for each chunk remains constant and does not increase as the video length grows, enabling real-time interactivity and enhancing the user's immersive experience. Furthermore, the first three rows of Fig. 13 illustrate the generated results spanning 637 frames.

### C.3. Comparison of Models under Context Forcing

We provide a comprehensive comparison of different models under context forcing in Table 8 and Fig. 16. The teacher model exhibits better control capability and visual quality due to the bidirectional nature, which provides

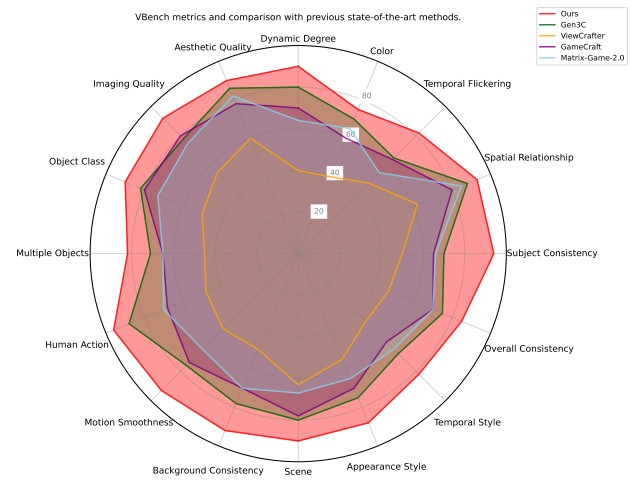

*Figure 15.* **VBench evaluation.**

reliable guidance during distillation. However, this limits its real-time interactivity. Through context forcing, we mitigate error accumulation while maintaining and even surpassing long-term consistency of the student model, yielding improved overall performance. In addition, context forcing reduces the student model's inference steps, enabling real-time interaction.

### C.4. Ablation for Memory Size

Table 8 evaluates the effect of different memory sizes. Using a larger spatial memory size leads to slightly better PSNR metric, while a larger temporal memory size better preserves the pretrained model's temporal continuity, resulting in better overall performance. Moreover, a larger spatial memory size may significantly increase the teacher model's memory size, as the spatial memory of adjacent chunks may completely differ, while their temporal memory overlaps. This not only increases the difficulty of training the teacher model but also poses challenges for distillation.

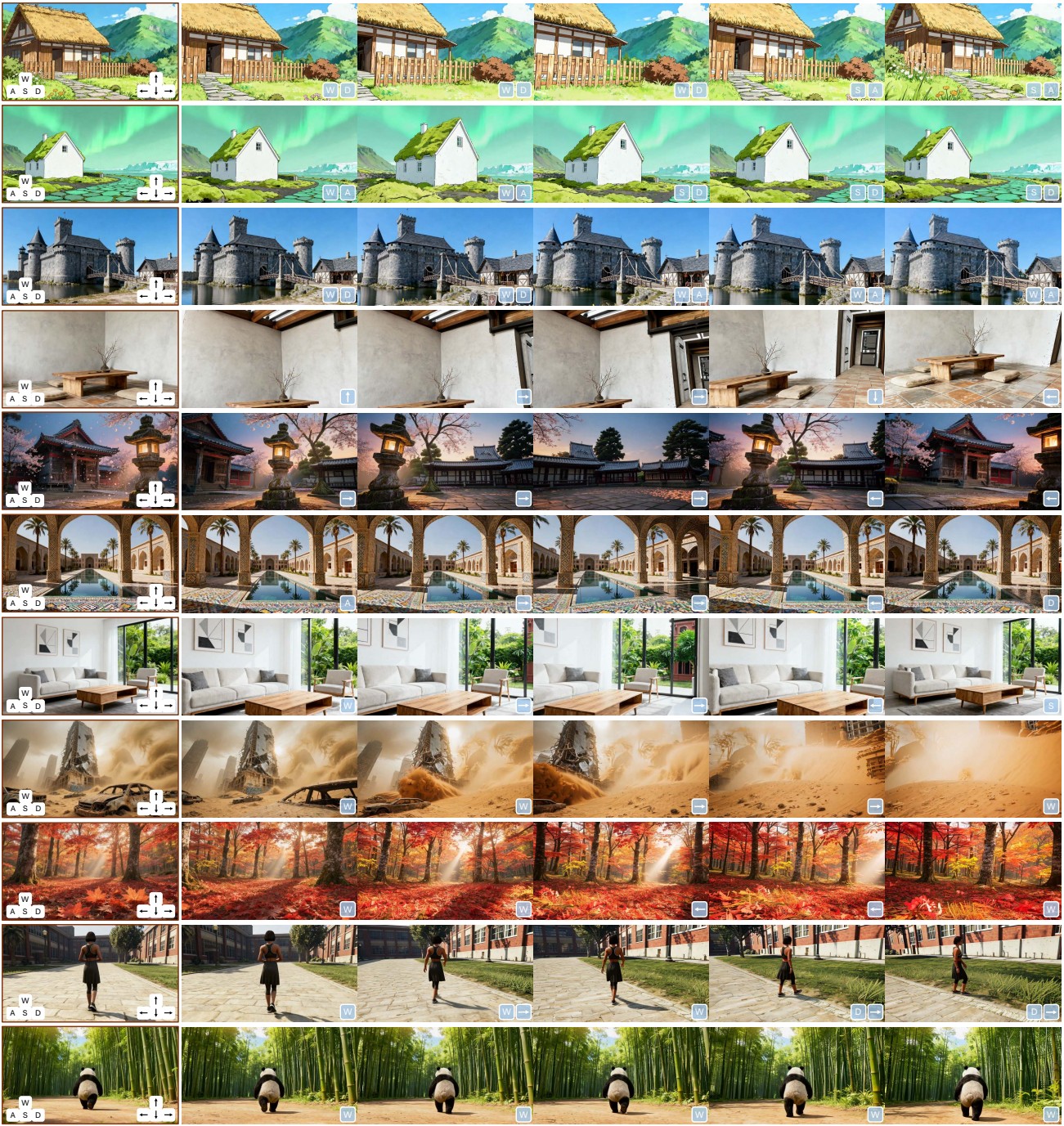

*Figure 12.* **More qualitative results.**

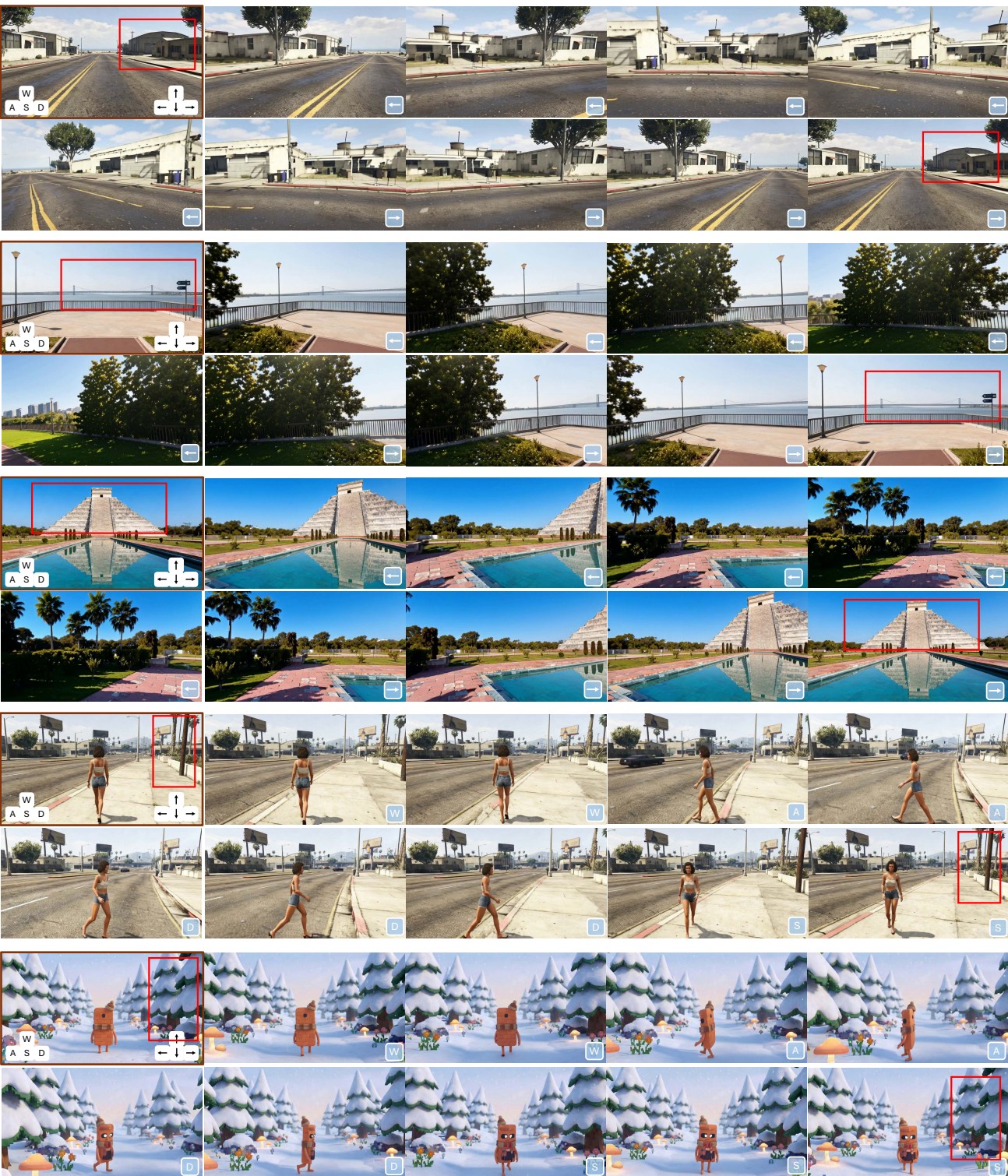

*Figure 13.* **More visualization results.**

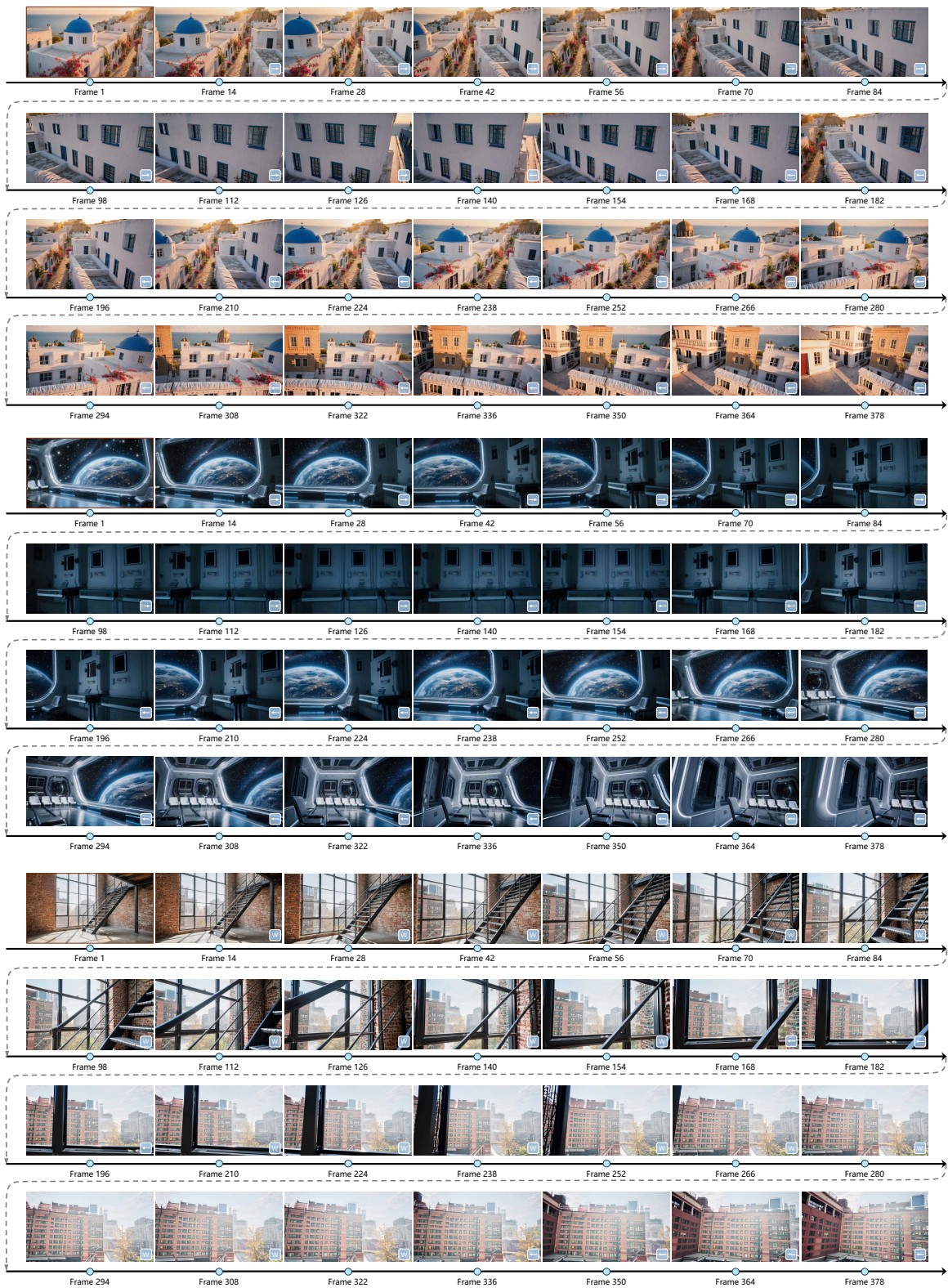

*Figure 14.* **Long video generation.**

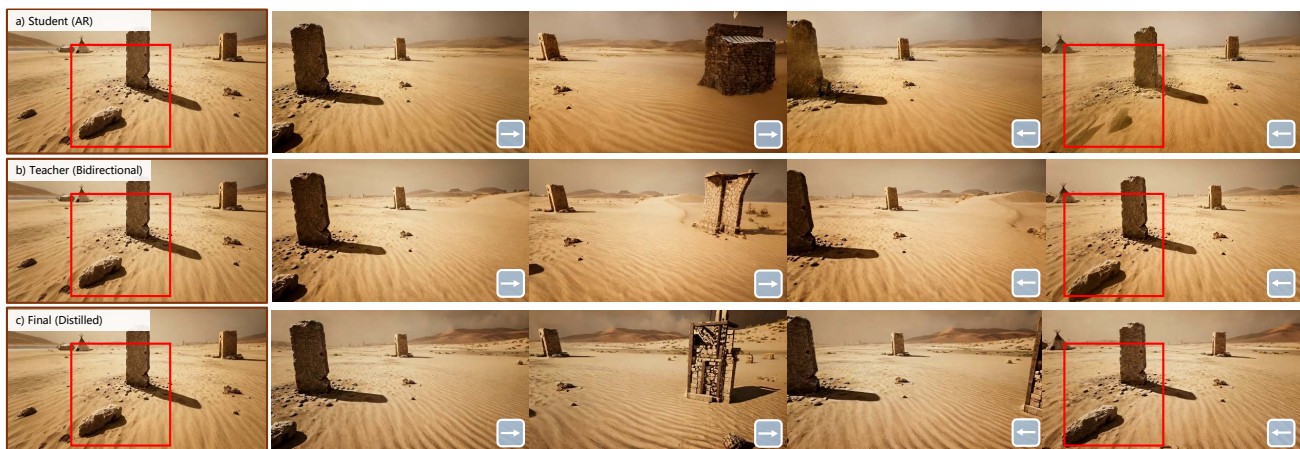

*Figure 16.* Visualization of different models under context forcing.

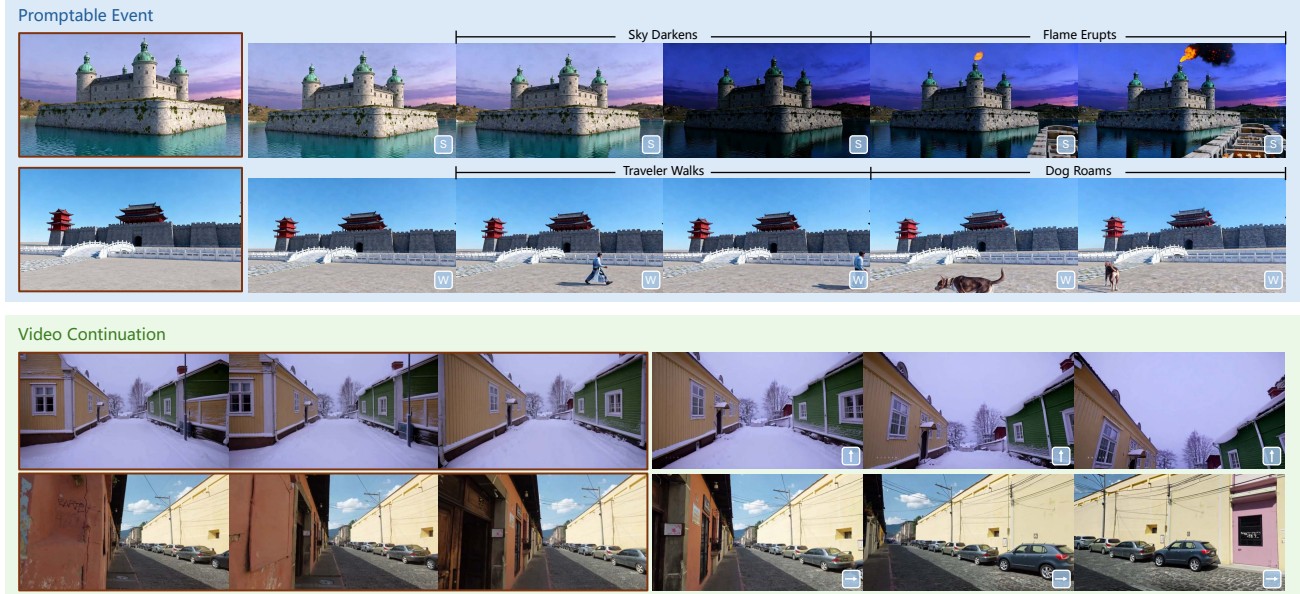

*Figure 17.* Visualization of promptable event and video continuation.

*Table 9.* Quantitative comparison on the *WorldScore*. **Bold and underline** presents the 1st, **Bold** indicates the 2nd, and underline means the 3rd.

| Method | WorldScore Average | Camera Control | Object Control | Content Alignment | 3D Consistency | Photometric Consistency | Style Consistency | Subjective Quality |
|---|---|---|---|---|---|---|---|---|
| WonderJourney (Yu et al., 2024) | 63.75 | 84.60 | 37.10 | 35.54 | 80.60 | 79.03 | 62.82 | 66.56 |
| WonderWorld (Yu et al., 2025a) | 72.69 | **92.98** | 51.76 | **71.25** | **86.87** | 85.56 | 70.57 | 49.81 |
| EasyAnimate (Xu et al., 2025) | 52.85 | 26.72 | 54.50 | 50.76 | 67.29 | 47.35 | 73.05 | 50.31 |
| Allegro (Zhou et al., 2024) | 55.31 | 24.84 | 57.47 | 51.48 | 70.50 | 69.89 | 65.60 | 47.41 |
| Gen-3 (Runway, 2024) | 60.71 | 29.47 | 62.92 | 50.49 | 68.31 | 87.09 | 62.82 | 63.85 |
| CogVideoX-I2V (Yang et al., 2024) | 62.15 | 38.27 | 40.07 | 36.73 | 86.21 | **88.12** | 83.22 | 62.44 |
| Voyager (Huang et al., 2025b) | **77.62** | 85.95 | **66.92** | 68.92 | 81.56 | 85.99 | **84.89** | **71.09** |
| **Ours** | **79.74** | **88.76** | **69.05** | 66.51 | **86.43** | **89.07** | **85.17** | 73.16 |

### C.5. Evaluation on VBench

We evaluate our model on VBench (Huang et al., 2024) across diverse metrics. For each baseline, we provide the same image and action to generate long-horizon videos. The results presented in Fig. 15 demonstrate the superior performance of WorldPlay. Notably, our method achieves outstanding results in key aspects such as consistency, motion smoothness, and scene generalizability.

### C.6. Evaluation on WorldScore

We conduct a comprehensive evaluation using the WorldScore (Duan et al., 2025) benchmark, which consists of 2,000 diverse test cases encompassing various styles and scenarios for the static setting. Each case requires generating content based on an input image, a text prompt, and a specific camera trajectory. To assess the effectiveness of our model, we compare it against several state-of-the-art 3D and video generation baselines. Following the evaluation protocol established by Voyager (Huang et al., 2025b), we focus on metrics that reflect controllability and generation quality across novel views. As demonstrated in Table 9, our method achieves the highest average score among all compared models, highlighting our model's superior performance in both precise controllability and high-fidelity generation quality.

## D. User Study

We conduct a comprehensive user study across multiple dimensions, including visual quality, control accuracy, and long-term consistency. In our setup, users are presented with two videos, generated from the same initial image and action inputs, and asked to select their preference based on the specified criteria. To ensure the robustness of our evaluation, we select 300 cases from diverse benchmarks such as VBench (Huang et al., 2024) and WorldScore (Duan et al., 2025), and 300 customized trajectories. The final results are then evaluated by a panel of 30 assessors. As shown in Fig. 18, compared to other baselines, our distilled model achieves superior generation quality across all aforementioned evaluation metrics, clearly demonstrating our model's capability for both real-time interaction and long-term consistency.

## E. Additional Applications

### E.1. Promptable Event

Due to the autoregressive nature of WorldPlay, we can modify the text prompt at any time to control the subsequent generated content. Specifically, inspired by LongLive (Yang et al., 2026), we employ a KV-recache technique to refresh the cached key–value states whenever the text prompt is modified. This effectively erases residual information from the previous prompt while preserving the motion and visual cues necessary to maintain temporal continuity. As shown in the upper part of Fig. 17, we can change the weather and trigger a fire eruption, or introduce new objects and characters. Through promptable event, we can generate various complex and

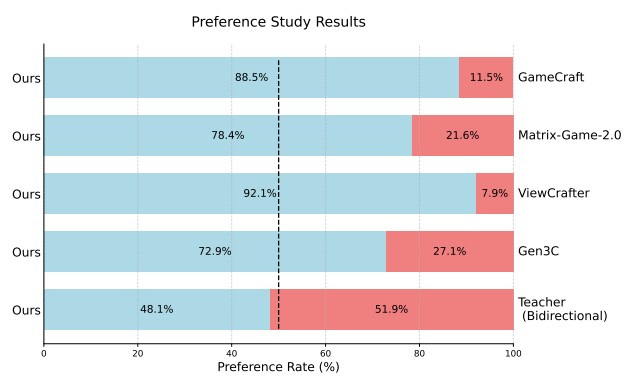

*Figure 18.* **Human evaluation.**

uncommon scenarios, which can benefit agent learning by enabling agents to handle these unexpected situations.

**E.2. Video Continuation**

As shown at the bottom of Fig. 17, WorldPlay can generate follow-up content that remains highly consistent with a given initial video clip in terms of motion, appearance, and lighting. This enables stable video continuation, effectively extending the original video while preserving spatial-temporal consistency and content coherence, which opens up new possibilities in creative video generation and virtual environment construction.

