# OpenReview forum: "WorldPlay: Towards Long-Term Geometric Consistency for Real-Time Interactive World Modeling"
_ICML.cc/2026/Conference — ICML 2026 regular_

### Official Review · Reviewer_hEP6 · 2026-03-11

**Soundness:** 4
**Presentation:** 3
**Significance:** 3
**Originality:** 2
**Overall Recommendation:** 5
**Confidence:** 5

**Summary:**

This paper presents WorldPlay, an autoregressive video diffusion-based world model for real-time streaming generation with long-term consistency. The method combines three main contribution: 1) a dual action representation fusing keyboard inputs + continuous camera poses, 2) a reconstituted context memory which rebuilds temporal/spatial memory and uses temporal reframing to fix RoPE indices, and 3) a context forcing distillation scheme that aligns teacher and student’s memory context to produce a long-term consistent real-time AR student. The paper’s experimental results validate its strength in practical application.

**Compliance With Llm Reviewing Policy:**

Affirmed.

**Final Justification:**

The paper tackles an important and timely problem in real-time interactive world modeling, and I find the overall method technically solid, with meaningful system-level contributions and strong empirical results. My main concerns were about clarity, positioning of novelty, and missing technical details; the rebuttal addressed these points well enough to reinforce rather than change my assessment, so I maintain my recommendation of 5 (Accept).

**Key Questions For Authors:**

- Could you explain how the teacher model is further trained to accept the memory cache? In general, I find it a bit difficult to estimate how the teacher model is trained to accept memory. It might be straightforward, but more descriptions, or perhaps a descriptive image should be helpful.
What’s the longest length that model can practically generate? Details such as how long videos are for stage 2 training, are missing so it’s hard to estimate the maximum length of video that model can generate.

- In the supplementary, it shows up to 378 frames but I am curious about the limits this memory approach can handle. Probably, it won’t be able to handle infinitely long sequences with consistent spatial/temporal memory – what’s the upper bound? Also, given that the system generates at 24 fps, 378 frames are roughly 15 seconds and may not be potentially enough to be claimed long-term.

**Limitations:**

The paper has a dedicated limitation section in the supplementary but is too general. Even proprietary state-of-the-art models like Genie 3 have certain limitations, and I recommend authors to discuss more fundamental/practical limitations of their approach.

**Strengths And Weaknesses:**

**Strength**
- This paper combines three non-trivial components, and I think they all technically make sense. Specifically, I think Context Forcing (while it cannot be easily adapted to T2V models as a memory-augmented teacher is required) could be a potentially great direction for world models with consistent memory.
- The paper discusses timely and potentially very important problems of world modeling. World modeling with autoregressive video diffusion models is a highly promising research direction that can be used in various fields such as robotics, game development, and interactive creative uses.
- As a researcher in this field, I highly value the paper’s engineering efforts to train models at this scale, with all of these techniques combined. Distillation process with DMD requires careful engineering (especially causal distillation w. roll-out at billion-scale dit backbone requires high level of distributed training and memory management) and serving parallel real-time inference in multiple GPUs is also not trivial. However, to better consider this as a contribution, I highly recommend authors to discuss more technical details adopted to serve this model or later open-source the implementation.

**Weaknesses**
- Writing is clear, but I think authors should tone-down and adjust tone (word choice) in overall writing. For instance, in line 45, authors mention “key innovations”, but I don’t think someone would call work as truly “innovative”. I understand the the authors’ engineering efforts (designing this large-scale model, and training it isn’t trivial) and agree that three components are non-trivial extensions of existing techniques. That said, dual action representation is a simple combination of existing 1) keyboard conditioning + PRoPE, 2) reconstituted context memory module is something new, but adjusting RoPE indices have been discussed (while not exactly the same) in previous works including [Rolling Forcing, MotionStream] (I also recommend citing these), and 3) a context forcing is something new, but not something I’d call innovative. Overall writing contains several components where it slightly overstates the work, although the work could already be valuable as is.
- Some basic details are missing such as base model size and how memory augmented teacher model is trained. Specifically, it’s a bit hard to imagine how context forcing is conducted when a memory-augmented bidirectional teacher model looks like.

---

> ### Author Rebuttal · Authors · 2026-03-30
>
> Thank you for recognizing and valuing our work. We address your constructive comments as follows:
>
> > **W1:** Writing is clear, but I think authors should tone-down and adjust tone......
>
> **A1:** We sincerely thank the reviewer for the recognition of our writing and efforts. Our initial use of the term 'key innovations' in the Abstract was intended to highlight the technical challenges in simultaneously achieving real-time generation speed and long-term geometric consistency within a world model. As stated by the reviewer, this task is non-trivial and extends beyond the simple integration of existing methods. However, we appreciate the reviewer's perspective on this terminology. In the revised version, we will replace 'key innovations' with 'key ingredients' (as used in the Introduction) and adjust the overall tone accordingly. Furthermore, we will include citations of RollingForcing and MotionStream.
>
> > **W2:** Some basic details are missing......
>
> **A2:** We apologize for the lack of clarity. Our teacher model is built upon the Wan-5B[1] and is implemented as a conditional bidirectional DiT. Following the training procedure in Supp. A, the teacher model is finetuned through a two-stage process. In the first stage, we first incorporate a dual-action representation into the bidirectional DiT and perform training to enable precise action control. Subsequently, as defined in Eq.5, we utilize $C_{j:j+3} - x_{j:j+3}$ as the teacher model’s memory context during the second stage of training, which yields the final teacher model employed for context forcing. We will include these model and training details in the revised manuscript.
>
> > **Q1:** Could you explain......
>
> **A3:** The teacher model is implemented as a conditional bidirectional DiT. During the training of stage two, the generation sequence consists of 4 chunks $x_{j:j+3}$ (16 latents, 61 frames). We utilize $C_{j:j+3} - x_{j:j+3}$ as a variable-length memory context for the teacher. Following the Flow Matching framework, we apply noise from $[0, 1]$ to $x_{j:j+3}$, while the memory context $C_{j:j+3} - x_{j:j+3}$ undergoes uniform noise sampling from $[0, 0.2]$ to enable robust conditioning. Crucially, the training loss are computed strictly on the generation sequence. In stage two, both the teacher and student models are trained on sequences of up to 160 latents (637 frames). While both teacher and student models can theoretically generate beyond this limit, we observe a gradual degradation in visual quality for much longer sequences. The results for 160-latent generation are shown in Fig.12 (first three cases). We will explicitly describe in the revised version to clarify the details.
>
> > **Q2:** In the supplementary......
>
> **A4:** This is a profound observation. As previously clarified (A3), both teacher and student models can theoretically generate beyond 160 latents. However, we observe a gradual degradation in visual quality for much longer sequences. Fig.12 (first three cases) presents long-term generation results of 637 frames, where we design revisit trajectories. As highlighted by the red boxes in the figure, the scene elements remain consistent when revisited, confirming that our model maintains long-term geometric consistency even over extended durations.
>
> > **Limitation:** Discuss more fundamental/practical limitations of their approach.
>
> **A5:**  We provide a more comprehensive discussion of the fundamental and practical limitations of our method. First, our model can generating videos of approximately 30 seconds, efficiently scaling this framework to longer durations [2,3] while maintaining long-term geometric consistency, remains a significant challenge. Second, although distillation is utilized to mitigate error accumulation, fundamentally averting this phenomenon during the training of diffusion models remains a critical open challenge [4]. Finally, retrieval mechanisms based on the FOV may fail to accurately identify memory context when faced with significant occlusions [5].
>
> [1] Wan5B. https://huggingface.co/Wan-AI/Wan2.2-TI2V-5B
>
> [2] Cui J, Wu J, Li M, et al. Self-forcing++: Towards minute-scale high-quality video generation[J]. arXiv preprint arXiv:2510.02283, 2025.
>
> [3] Cui J, Wu J, Li M, et al. LoL: Longer than Longer, Scaling Video Generation to Hour[J]. arXiv preprint arXiv:2601.16914, 2026.
>
> [4] Guo Y, Yang C, He H, et al. End-to-end training for autoregressive video diffusion via self-resampling[J]. arXiv preprint arXiv:2512.15702, 2025.
>
> [5] Yu J, Bai J, Qin Y, et al. Context as memory: Scene-consistent interactive long video generation with memory retrieval[C]. SIGGRAPH Asia. 2025: 1-11.

---

> > ### Author Rebuttal · Reviewer_hEP6 · 2026-04-02
> >
> > Most of my concerns have been adequately addressed. I will maintain my rating of 5 (accept). I encourage the authors to incorporate the discussed points into the final manuscript. Great work!

---

> > > ### Author Response · Authors · 2026-04-02
> > >
> > > Thanks for your recognition and positive feedback. We will incorporate the discussed points into the final manuscript.

---

### Official Review · Reviewer_d1JE · 2026-03-12

**Soundness:** 4
**Presentation:** 4
**Significance:** 3
**Originality:** 4
**Overall Recommendation:** 5
**Confidence:** 4

**Summary:**

This paper proposed Worldplay, a real-time interactive world model supporting both first-view and thrid-view explorations. Three novel contributions are proposed: A Dual Action representation is proposed to combine both continuous signals with discrete signals; A Reconstituted Memory module with temporal reframing is designed to guarantee long-term consistency; A context forcing model is introduced to align teacher and student networks, enabling high efficiency.

**Compliance With Llm Reviewing Policy:**

Affirmed.

**Final Justification:**

My concerns are addressed, and I keep my acceptance score.

**Key Questions For Authors:**

I have some questions which are expected to further discuss in the rebuttal:

1) In Figure 4, it is a little hard to comprehend the temporal index. Could you describe more about this figure's meaning?

2) As for Context Forcing, there is one paper proposed later [1]. Could authors compare the difference with this one?

3) In the evaluation metrices, more explanations in R_{dis} T_{dis} could further strengthen the understanding about cumulation error. Are these for rotation and translation errors when revisiting the same place?

[1] Context Forcing: Consistent Autoregressive Video Generation with Long Context

**Limitations:**

Authors didn't give the limitation part about this work.

**Strengths And Weaknesses:**

Strengths:

1. The paper tackles three crucial challenges in interactive world model field, which offers great insight and effectively facilitate subsequent research.

2. The novelty of three core modules are good and effective. The description and motivation are also strong backuped.

3. Compared to prior methods, this paper can guarantee long-term consistency, especially when revisiting the same scenarios.

4. The overall writing is good and authors can learn a lot, including recent progress in interactive world model, video diffusion models, etc, from reading this paper.

Weaknesses:
None

---

> ### Author Rebuttal · Authors · 2026-03-30
>
> Thank you for recognizing and valuing our work. We address your constructive comments as follows:
>
> > **Q1:** In Figure 4, it is a little hard to comprehend the temporal index......
>
> **A1:** Current video diffusion models employ 3D RoPE, where the temporal indices are assigned based on the absolute sequential order of latents, as illustrated in Figure 4(a). To reduce computational overhead, our Reconstituted Context Memory selects a subset of chunks as memory context (e.g., in Figure 4(b), the 8th chunk selects chunks 2, 3, 6, and 7 as memory context). However, using absolute temporal indices (2, 3, 6, 7, 8) introduces two challenges: 1) The increasing relative distance (e.g., $8 - 2 = 6$) may exceed the trained interpolation range of RoPE. 2) The decay mechanism in RoPE weakens the influence of long-past spatial memory on current chunk as the relative distance grows. To resolve this, we propose Temporal Reframing (Figure 4(c)), which re-assigns relative indices (0, 1, 2, 3, 4) to the selected context (chunks 2, 3, 6, 7, and 8). This operation effectively “pulls” important past frames closer in time (the relative distance between 8th chunk and 2th chunk is $4-0=4$) and maintains the maximum relative distance ($4-0=4$) within the trained interpolation range of RoPE, ensuring their sustained influence and enabling robust extrapolation for long-term consistency.
>
> > **Q2:** As for Context Forcing, there is one paper proposed later......
>
> **A2:** The fundamental distinction between our method and Context Forcing lies in the mechanism of memory construction. Context Forcing categorizes memory into Attention Sink, Slow Memory, and Fast Memory, we diverge significantly in the selection strategy for Slow Memory. Our method employs a geometric relevance-based sampling, which is tailored for preserving long-term geometric consistency in world model, which is often overlooked by similarity-based metric used in Context Forcing. Consistent with the findings in Context Forcing, we also observed the empirical effectiveness of Attention Sink (as detailed in Supp. A). Additionally, our Temporal Memory shares a similar logic with their Fast Memory to maintain temporal consistency.
>
> > **Q3:** In the evaluation metrices......
>
> **A3:** We wish to clarify that in both short-term and long-term experiments, $R_{dis}$ and $T_{dis}$ are calculated on the entire trajectories. The metrics are specifically designed to evaluate the accuracy of action control, rather than to measure error accumulation. In the context of our paper, 'error accumulation' refers to the phenomenon where video generation quality degrades as the sequence length increases. This is quantitatively evidenced by the PSNR, SSIM, and LPIPS metrics in Table 2. Ours (full) achieves consistently superior performance compared to Ours (w/o Context Forcing), especially in long-term scenarios. This demonstrates that Context Forcing effectively mitigates error accumulation, thereby maintaining high visual fidelity over long sequences. We are not entirely sure if we misunderstood your comment. If we have misunderstood, please feel free to point it out so we can further clarify.
>
> For limitations, please refer to A5 in Review hEP6.

---

> > ### Author Rebuttal · Reviewer_d1JE · 2026-04-01
> >
> > Thanks authors for detailed explanations. Most of my concerns are addressed. However, I still have doubts about how $R_{dis}$ and $T_{dis}$ are calculated to indicate action control. More formulas and benchmarking details are helpful for understanding. Overall, this is a great work.

---

> > > ### Author Response · Authors · 2026-04-02
> > >
> > > We sincerely thank the reviewer for the positive feedback. Following your suggestion, we provide a more detailed clarification regarding the formulas and benchmarking details, which will be added into the revised version. $R_{dis}$ and $T_{dis}$ are designed to evaluate the alignment between the camera poses of the generated videos and the ground truth poses. Specifically, we use the ground truth poses as the input for each model to generate videos. Then, we utilize ViPE to estimate the camera poses of the generated videos. Following the previous works [1,2], we compute the relative poses of the ground truth and generated camera poses by setting the extrinsic matrix of first frame as an identity matrix and normalize the translation scale using the furthest frame. The rotation distance $R_{\text{dis}}$ is calculated by comparing the predicted rotation matrices $R_{gen}$ and ground truth rotation matrices $R_{gt}$:
> > > $$
> > > \begin{aligned}
> > > R_{\text{dis}}= \text{arccos} (\frac{\text{tr}(R_{\text{gen}} R_{\text{gt}}^{\top}) - 1}{2})
> > > \end{aligned}
> > > $$
> > > where $\text{tr}(\cdot)$ denotes the trace of a matrix. The translation distance $T_{dis}$ is computed by comparing the predicted $t_{\text{gen}}$  and ground truth translation vectors $t_{\text{gt}}$:
> > > $$
> > > \begin{align}
> > > T_{\text{dis}} = || t_{\text{gt}} - t_{\text{gen}} ||_2
> > > \end{align}
> > > $$
> > >
> > > [1] He H, Xu Y, Guo Y, et al. Cameractrl: Enabling camera control for text-to-video generation[C]. ICLR. 2025.
> > >
> > > [2] Yu W, Xing J, Yuan L, et al. Viewcrafter: Taming video diffusion models for high-fidelity novel view synthesis[J]. TPAMI. 2025.

---

### Official Review · Reviewer_ZmnJ · 2026-03-12

**Soundness:** 3
**Presentation:** 2
**Significance:** 4
**Originality:** 3
**Overall Recommendation:** 4
**Confidence:** 4

**Summary:**

This paper presents WorldPlay, a new video diffusion (world) model that addresses limitations of existing models. Specifically, the key issue that is being tackled in long-term consistency in the autoregressive generation.

The paper introduces three main novelties: (1) using both the action (e.g., keystroke) and resulting camera parameters (extrinsics/intrinsics) as inputs to the model as the action conditioning, (2) a context memory system that chooses to keep recent frames but also past frames that are spatially promixal (and re-indexing the frames so that RoPE does not have to extrapolate past its trained distribution), and (3) a method called "Context Forcing" that modifies the teacher model so that the latent distributions of the teacher and student models are both conditioned on the same past context, which juxtaposes previous approaches where the teacher model is bidirectional and global.

Experimental results show that WorldPlay achieves SOTA quantitative performance in both short-term and long-term generation. The paper also presents a few qualitative examples of improved long-term consistency and memory.

**Compliance With Llm Reviewing Policy:**

Affirmed.

**Final Justification:**

My concerns are address, I maintain my opinion to accept this paper.

**Key Questions For Authors:**

While this paper presents impressive results with seemingly sufficient research-based novelty, I have some concerns about the lack of clarity in the manuscript, which prevents me from giving a higher score at this time. In addition to the questions posed in the "Weaknesses" section, I have a few other minor clarification questions:

1. In Section 3.1, it is stated that current video diffusion models typically consist of a 3D VAE. However, in Supp. Mat, the authors claim to use a 4D VAE. If that is true, that is never mentioned in the main paper, and it would seem logical that this paper also use a 3D VAE since it is a "current video diffusion model". Please clarify this point with precision.

2. What exactly do you mean by augmenting the teacher model with memory and "structuring its context"? Are you saying that the teacher model is also now a causal DiT? Or are there multiple teacher models being trained? In general, please elaborate on the Context Forcing algorithm with more rigor and careful detail.

3. Is the "RoPE design" ablation study referring to the "reconstituted context memory" in Section 3.3? While I am almost sure this is the case, it is a bit ambiguous since the terminology used in Section 3.3 and Section 4.2 "RoPE Design." is different and there is no cross-referencing between the two portions of text.

**Limitations:**

Future directions are mentioned, but not limitations. Please briefly address this.

**Strengths And Weaknesses:**

S1. In an obviously relevant area of research, this paper presents what seems to be a state-of-the-art model, where performance gains are not simply attributable to data and scale, but to simple yet effective novelties in architecture and training. I would like to hear from other reviewers whether the baseline methods presented in the paper are fully representative of the current literature, as I know there is much work being done in this field.

S2. The qualitative examples provided in the paper are persuasive and provide useful complementary information to the quantitative results. In particular, Figures 6,7, and 9, as well as Figure 4 which provides an easy-to-understand overview of the "reconstituted context memory" idea.

S3. The paper is easy to follow and well-structured.

W1. In terms of the dual action representation, I don't fundamentally see the argument for why it makes sense to use both discrete keys and camera parameters other than that there is more information provided. At the very least, the argument is not rigorous, and could be attributed to the rather short textual description of this novelty (Sec 3.2). Furthermore, Figure 3 is quite difficult to understand; for instance, what does the arrow from the "continuous camera tokens" to the (D^proj)^{-1} symbol mean exactly? Please provide a much clearer and extensive explanation of *what* the pipeline looks like, and theoretically *why* the design choices make sense (e.g., why not concatenate both the Discrete Key Tokens and Continuous Camera Tokens to the time embedding). This would play a significant role in my final evaluation of the paper.

W2. Along the same lines, in terms of the reconstituted context memory, what does it mean that the sampling is guided by "geometric relevance scores"? Please elaborate on this and justify the design choice. Furthermore, while I do understand that the dynamic re-assignment would improve over simply no re-assignment, did the authors experiment with a re-assignment based on not only temporal, but also spatial proximity?

W3. Along the same lines, in terms of the context forcing, the idea of "self-rollout" is not explained (at all), which hinders my understanding of the context forcing process (although I do still get the gist). What is "self-rollout"? Please explain this more carefully.

W4. There are no quantitative experiments to justify the design choices for real-time latency presented in Section 3.5. Please provide ablation studies, or similar experimental results.

W5. While I appreciate the results for downstream applications (Section 4.3), especially promptable events shown in Figs. 9 and 1(e), I am confused about what it means to "integrate a 3D reconstruction model" for the 3D reconstruction downstream task. Please provide details.

W6. Figure 1, while visually appealing, does not in my opinion provide any insight into the pipeline with any amount of clarity. For starters, the "10" and "20" seem quite random, and I am not sure what copy and pasting basically two copies of the same figure side-by-side is supposed to show me. Please elaborate on what I am supposed to learn from this figure.

I also have a few more minor clarification questions; please see "Key Questions for Authors" section.

---

> ### Author Rebuttal · Authors · 2026-03-30
>
> Thank you for recognizing and valuing our work. We address your constructive comments as follows:
>
> > **W1:** In terms of the dual action representation......
>
> **A1:** While camera poses are informative, obtaining scale-consistent poses for diverse videos is challenging due to the scale ambiguity in existing pose estimation models. Training on inconsistent scales often leads to instability. Therefore, we utilize discrete keys to bypass the scale inconsistency, providing scale-invariant control signal that generalizes across different scenes. As shown in Tab.3, our dual-action representation achieves superior performance.
>
> In Fig.3, $D_{proj}^{-1}$ is the inverse matrix derived from intrinsic $K$ and extrinsic $T^{cw}$, which is compute as follows:
> $$
> D_{proj} = \begin{bmatrix}
> K & 0 \\\\
> 0 & 1
> \end{bmatrix} T^{cw}
> $$
> We encode discrete keys using PE and an MLP, adding them to the timestep embedding (Fig.3, left). For camera tokens, we compute $D_{proj}$ and inject it via the red path (Eq.3), while preserving the original self-attention along the blue path (Fig. 3, right). Unlike naive concatenation, PRoPE offers two advantages for spatial reasoning: 1) it explicitly encodes dense viewpoint relationships between frames, rather than treating them as isolated vectors. 2) as a relative position encoding, it makes the model invariant to the arbitrary definition of the world frame.
>
> > **W2:** In terms of the reconstituted context memory......
>
> **A2:** Our geometric relevance score is derived from FOV overlap and camera distance. Following WorldMem, we use a Monte Carlo method to compute overlap ratios. If overlap ratios are tied, we prioritize the pose closest to the current pose. The score is used to sample key historical frames, reducing computational overhead.
>
> We appreciate the reviewer's insightful observation. While re-assign index based on spatial proximity might further "pull" important past frames closer, it may introduce index discontinuity between the current frame $t$ and its temporal predecessor $t-1$, thereby undermining the temporal consistency. Moreover, it may assign a smaller temporal index to a frame relative to its predecessor. This index inversion violates the intrinsic causal priors and monotonic temporal expectations acquired by the pretrained VDM, leading to training instability and degraded generation quality.
>
> > **W3:** In terms of the context forcing......
>
> **A3:** In Context Forcing, the student model utilizes its previously generated chunks as the context for subsequent chunks, this process is defined as self-rollout.
>
> > **W4:** There are no quantitative......
>
> **A4:** We conduct quantitative experiments in the table below. This will be added into the revised version.
>
> | DiT P&Q | VAE P&Q | Streaming Decoding | FPS |
> | :---: | :---: | :---: | :---: |
> | | | | 2.80 |
> | ✓ | | | 3.80 |
> | ✓ | ✓ | | 16.0 |
> | ✓ | ✓ | ✓ | 24.0 |
>
> *"P" and "Q" denote Parallelism and Quantization.*
>
> > **W5:** While I appreciate......
>
> **A5:** For the 3D reconstruction applications, we first utilize our model to autoregressively generate a video. This video is then processed by a 3D reconstruction model to produce the final point clouds.
>
> > **W6:** Figure 1, while visually appealing......
>
> **A6:** We first clarify that the reviewer likely refers to Fig.2. Fig.2 provides an overview of our real-time interactive world model. Here, '10' and '20' indices denote the streaming process, while the 'User Input' and 'Memory Cache' highlight interactivity and long-term consistency, respectively. The lightning icon signifies real-time. We selected a revisit trajectory as the example to underscore the long-term consistency. The identical spatial contexts ($C_{10}^S = C_{20}^S$) demonstrate that we may retrieval the same frames during revisit trajectory.
>
> > **Q1:** In Section 3.1......
>
> **A7:** We could not find '4D VAE' in our Supp. Could you please clarify the specific lines where this term appears? We could provide explanation once the location is identified.
>
> > **Q2:** What exactly......
>
> **A8:** "Augmenting the teacher model with memory" presents supervised fine-tuning of a single, conditional bidirectional DiT to achieve long-term geometric consistency. For details, please refer to A3 in Review hEP6. As shown in Algo.1, Context Forcing establishes a distillation between the teacher and the student model, which begins with the student self-rollout to generate a video sequence. This sequence is then fed into the frozen teacher and a trainable "fake" model (initialized from the teacher) to compute real and fake scores, respectively. We optimize the student via the DMD loss (Eq.4), while updating the fake model using a Flow Matching loss on the self-rollout sequence.
>
> > **Q3:** Is the "RoPE design"......
>
> **A9:** As you correctly identified, the 'RoPE design' is refer to 'reconstituted context memory'. We will incorporate explicit cross-references in the revised version.
>
> For limitations, please refer to A5 in Review hEP6.

---

> > ### Author Rebuttal · Reviewer_ZmnJ · 2026-04-04
> >
> > Thank you for the clarifications. I have no further questions at this time.

---

> > > ### Author Response · Authors · 2026-04-04
> > >
> > > Thank you for reviewing our rebuttal. We are pleased that our responses have fully resolved your questions.

---

### Official Review · Reviewer_WprB · 2026-03-13

**Soundness:** 4
**Presentation:** 4
**Significance:** 3
**Originality:** 3
**Overall Recommendation:** 4
**Confidence:** 4

**Summary:**

The paper focused on the task og long-term video-gen-based world modeling, and proposed WorldPlay, an novel framework that integrates the technique of Dual Action Representation, Reconstituted Context Memory, and an improved self-forcing method. The experiments show that the proposed method can achieve SoTA performance.

**Compliance With Llm Reviewing Policy:**

Affirmed.

**Final Justification:**

The authors have addressed all my concerns. I will maintain my positive score.

**Key Questions For Authors:**

Please see the weaknesses section

**Limitations:**

yes

**Strengths And Weaknesses:**

### Strengths
- The presentation of figures is great in this paper. The paper is self-contained and easy-to-follow.
- The experiments are comprehensive and show the superior performance of the proposed method.
- The discussion and improvement about the balance of consistency and efficiency is interesting.
- The idea of Reconstituted Context Memory is straightforward and technically sound (especially the trick of Temporal Reframing).

### Weaknesses
- Spatial memory sampling is scored based on factors such as FOV overlap and camera distance. If pose estimation is noisy, the retrieval process may select the wrong frame, leading to consistency errors or artifacts.
- The metrics primarily rely on image similarity and motion error along the revisiting trajectories. They may still fail to detect frames that merely resemble the original in appearance but have inconsistent 3D structures.

---

> ### Author Rebuttal · Authors · 2026-03-30
>
> Thank you for appreciating and acknowledging our work. We address your constructive comments below:
>
> > **W1:** Spatial memory sampling is scored based on factors such as FOV overlap and camera distance......
>
> **A1:** We agree that pose estimation noise may negatively impact spatial memory retrieval and consistency. We address this concern from two perspectives: 1) As detailed in Supp. B, we also utilize real-world 3D scene data and UE rendering data with exact, ground-truth camera poses to train our model. Unlike reconstruction-based pose estimation (e.g., ViPE), these datasets allow us to extract camera poses directly, which effectively eliminates estimation error, ensuring precise retrieval and robust spatial consistency. 2) We also implement an automated filtering process to detect and remove video clips exhibiting erratic camera transitions and rotation angles. This preprocessing step further guarantees data quality, thereby mitigating the risks associated with retrieving wrong frames.
>
> > **W2:** The metrics primarily rely on image similarity......
>
> **A2:** We thank the reviewer for the suggestion. To rigorously evaluate the long-horizon 3D structural consistency of our model, we have incorporated a more advanced metric, MEt3R [1], which explicitly models the multi-view correspondence by leveraging pre-trained 3D geometric priors (e.g., DUSt3R). Specifically, we evaluate the generated long-horizon videos by feeding corresponding frame pairs from the initial and revisiting trajectories into the model to measure the 3D structural alignment. The results are summarized in the table below, which clearly demonstrates the superiority of our approach in maintaining 3D consistency.
>
> | Method | MEt3R Score ($\downarrow$) |
> | :--- | :---: |
> | Gen3C | 0.188 |
> | Matrix-Game-2.0 | 0.367 |
> | GameCraft | 0.305 |
> | **Ours** | **0.133** |
>
> [1] Asim M, Wewer C, Wimmer T, et al. Met3r: Measuring multi-view consistency in generated images[C]. CVPR. 2025: 6034-6044.

---

> > ### Author Rebuttal · Reviewer_WprB · 2026-04-02
> >
> > Thanks for the response.
> > I have three more questions:
> > - Could you detail the "automated filtering process" you mentioned in A1? How does it filter the unacceptable data?
> > - For A1, it seems that you solved this issue in a data-driven manner. How do you consider overcoming this issue in your method design? I am curious about whether it is enough to address this issue just by improving the training data. The noisy pose estimation might still exist.
> > - The paper claimed that "our method supports text-based manipulation during streaming." What kind of manipulation could your method trigger? Could you show some video demos for this? Only a grid of video frames might not be enough to evaluate overall performance.

---

> > > ### Author Response · Authors · 2026-04-03
> > >
> > > Thanks for your time for reviewing our rebuttal. We address your more questions as follows:
> > >
> > > > **MQ1:** Could you detail the "automated filtering process"......
> > >
> > > **MA1:** For unannotated video clips, we employ ViPE for camera pose estimation. To ensure the temporal smoothness of the predicted poses and eliminate estimation failures, we utilize the Peak-to-Median Ratio (PMR, the ratio of the maximum inter-frame motion to the median) of inter-frame motions as a detection metric. Specifically, we subsample the predicted poses every four frames and compute the relative transformations to approximate the camera's instantaneous velocity. By evaluating the PMR, we can robustly identify impulsive pose jumps. Clips exhibiting a PMR above a conservative threshold (set to 5.0 in our experiments) are classified as unacceptable and are discarded.
> > >
> > > > **MQ2:** For A1, it seems that you solved this issue in a data-driven manner......
> > >
> > > **MA2:** For data that may still contain pose noise, our model demonstrates robustness by leveraging implicit conditioning and attention mechanisms, which enable it to learn to attend to relevant information while ignoring irrelevant context. While enhancing data quality remains a vital objective, developing more robust retrieval mechanisms and architectures also represents another important research direction, which we will add into our future work.
> > >
> > > > **MQ3:** The paper claimed that......
> > >
> > > **MA3:** Our method supports the text-based manipulation of **"environment changes"** and
> > > **"object appears"**. Representative examples of "environment changes" are shown in our supplementary video (from 65s to 83s), while additional demonstrations of "object appears" are accessible via https://anonymous.4open.science/api/repo/ICML_prompt_event-33C0/file/Object_appear.mp4?v=096f3d08. We will add the details into the revised version.

---

### Decision · Program_Chairs · 2026-04-30

**Decision:**

Accept (regular)

**Comment:**

I recommend Accept (medium priority: solid contribution to ICML). The reviewer consensus is clearly positive, with no major disagreement. Reviewers generally agree that this paper tackles an important and timely problem in real-time interactive world modeling, and that the proposed system is technically solid. In particular, they find the combination of dual action representation, reconstituted context memory, and context forcing well motivated, and view the empirical results on long-term consistency, control, and real-time generation as strong. Several reviewers also highlighted the paper’s practical relevance and the nontrivial engineering effort required to make this system work at scale.

The main concerns were mostly about clarity and completeness rather than the core contribution itself. Reviewers asked for more explanation of several design choices, including the dual action representation, the memory retrieval and temporal reframing mechanism, the context forcing procedure, and some evaluation details such as pose-based metrics, latency, and system limits. One reviewer also raised concerns about robustness to noisy pose estimation and whether the evaluation fully captures 3D consistency. In the rebuttal, the authors provided detailed clarifications, added further quantitative evidence, and pointed to additional demos and analysis. These responses appear to have addressed the substantive concerns, and the final reviewer feedback remained uniformly positive.

Overall, I find this to be a strong systems-oriented paper on a fast-moving topic, with solid technical contributions and convincing experimental validation. While the paper would benefit from incorporating the clarified technical details more directly into the final version, the current submission already makes a meaningful contribution that should be of interest to the ICML community.